# In-depth single molecule localization microscopy using adaptive optics and single objective light-sheet microscopy

Marine Cabillic[1,2,9], Hisham Forriere[1,9], Laetitia Bettarel[1], Corey Butler [1,3], Abdelghani Neuhaus [1], Ihssane Idrissi[1,2], Miguel Edouardo Sambrano-Lopez[1], Julian Rossbroich [1], Lucas-Raphael Müller[4], Jonas Ries [4,5], Gianluca Grenci [6,7], Virgile Viasnoff [6], Florian Levet [1,8], Jean-Baptiste Sibarita [1] ✉ & Rémi Galland [1] ✉

Single molecule localization microscopy (SMLM) allows deciphering the nanoscale organization and dynamics of biomolecules in their native environment with unprecedented resolution. While SMLM was quickly adopted by the scientific community for its performance and simple instrumentation, it still remains limited in its in-depth capability, precluding many biological processes to be investigated. We here present a solution to perform in-depth volumetric SMLM, called soSMARt. It relies on innovative microfabricated devices allowing both single-objective light-sheet microscopy, aberrations correction via adaptive optics, and real-time feedback-loop registration with nanometric precision. We illustrated the performances of soSMARt to assess the 3D nanoscale organization of several protein of interest in isolated cells, and explore optimizations and proof-of-concepts for the investigation of more complex tissues such as 3D cell cultures. We believe our method addresses key limitations of single molecule microscopy, paving the way for novel biological applications.

The advent of Single Molecule Localization Microscopy (SMLM) has drastically improved our capability to assess the organization and dynamics of proteins of interest within their native environment, with unprecedented spatial resolution, leading to many new discoveries in biology, neuroscience, and fundamental and applied research[1,2]. SMLM relies on the sparse activation of isolated fluorophores followed by their precise localization by analytical methods[3]. Several methods now make it possible to localize single molecules in 3D, including optical shaping of the point spread function (PSF) to discriminate the axial position of the single emitters. The final super-resolution image or volume is reconstructed by accumulating all the localizations over time, and its resolution mostly depends on the number of detected photons per localization event as compared to the fluorescence background, the sharpness of the PSF, and the density of localized molecules[4]. Best resolutions are therefore achieved using microscopy techniques with high numerical aperture (NA) objectives, limited optical aberrations, and optimal optical sectioning, making in-depth 3D SMLM challenging. As a consequence, the most popular illumination strategies for SMLM are TIRF and HILO[5], as they feature both a high signal to noise ratio with very low background with limited optical

¹Univ. Bordeaux, CNRS, IINS, UMR 5297, FR-33000 Bordeaux, France. ²Sanofi, Integrated Drug Discovery Department, Vitry-sur-Seine, Paris, France. ³Imagine Optic, Orsay, France. ⁴Cell Biology and Biophysics Unit, European Molecular Biology Laboratory (EMBL), Heidelberg, Germany. ⁵Max Perutz Labs, Department of Structural and Computational Biology, University of Vienna, Vienna, Austria. ⁶Mechanobiology Institute, National University of Singapore, Singapore, Singapore. ⁷Biomedical Engineering Department, National University of Singapore, Singapore, Singapore. ⁸Univ. Bordeaux, CNRS, INSERM, BIC, US 4, UAR3420, FR-33000 Bordeaux, France. ⁹These authors contributed equally: Marine Cabillic, Hisham Forriere.
✉e-mail: jean-baptiste.sibarita@u-bordeaux.fr; remi.galland@u-bordeaux.fr

aberrations. However, they restrict the observation of biomolecules at the close proximity of the glass coverslip, inherently preventing the possibility to investigate biological events at the whole cell scale or deeper within tissues.

Light-Sheet Fluorescence Microscopy (LSFM) technics, which is mostly popular for imaging thick biological samples[6,7], have been demonstrated already 20 years ago to be suited to image single molecule in depth[8]. On its conventional implementation, LSFM relies on the focalization, and potentially scanning, of a Gaussian beam through a first objective to form a thin excitation light-sheet located at the focal plane of a second objective positioned perpendicularly (90°) to the first one. However, the mechanical constraints of such multi-objective configuration limit the NA of the collection objective and consequently the achievable localization performances in SMLM.

Over the past years, several strategies have been proposed to overcome this limit, allowing using high NA detection objectives in a LSFM configuration. Most of them rely on increasing the angle in between the excitation and collection objectives, in order to reduce the mechanical constraints of conventional 90° architectures. It was initially performed by deflecting the illumination light-sheet using a prism[9,10], a mirror[11,12], or simply using a tilted light-sheet according to the image plane[13]. Another approach was to specifically design an excitation objective whose dimensions allow to use a high NA collection objective[14–16]. Finally, single objective-based LSFM architecture were also proposed to perform SMLM several microns above the coverslip. It uses either an oblique light-sheet combined with a remote detection system to project the tilted excited area onto the camera[17,18], or a custom made sample holder integrating mirrors to create, after reflection, a light-sheet perpendicular[19,20], or slightly tilted[21] to the optical axis of the objective.

By seeking to perform SMLM at the whole cell scale, those systems endeavored to address some important challenges. The first challenge concerns mechanical drift correction during acquisition. In a single plane acquisition, this is usually achieved offline, using either fiduciary markers adsorbed onto the glass coverslips or analytical approaches such as cross-correlation[22]. However, positioning fiduciaries in depth at different focal planes is not straightforward, and the long acquisition time needed to images entire volumes by SMLM usually requires real-time correction. This can be addressed using brightfield-based cross-correlation[23] or by localizing non-fluorescent beads over extended depth range[24], but at the cost of an additional detection channel. A second challenge concerns the reconstruction of the entire volume acquired at the nanoscale, which depends on the acquisition strategy. In the case of plane-by-plane sequential acquisitions, adjacent plane reconstructions need to be registered and stitched, whereas for volume-by-volume acquisition, each volume must be registered to the others. Finally, a third challenge is optical aberrations, which increase with imaging depth and reduce localization accuracy. This is particularly problematic for PSF shaping-based 3D localization approaches[25,26], as they are extremely sensitive to optical aberrations[27,28]. While optical aberrations can be neglected when imaging cells spread onto the glass coverslips, they become prominent when imaging non-adherent cells located more than 10 µm above the coverslip, or thick multi-cellular samples.

Several solutions have been proposed in the literature to address all, or part of these issues (Supplementary Fig. 1). One approach involves simultaneously acquiring a whole volume, up to 8 µm in depth, using techniques such as Multi-Focus[29,30] or Light-Field Microscopy[31,32], or long-axial range PSF[33]. They allow for the simultaneous detection of single molecules throughout an extended volume, facilitating their relative repositioning. They also enable the detection of fiduciary markers located on the coverslip for real time or offline drift correction. However, these solutions lack in-depth optical sectioning and do not account for optical aberrations, limiting their applicability to deeper imaging. Another approach involves

repetitively acquiring a volume that includes at least one fiduciary marker located onto the coverslip, which allows for its registration[14,16]. However, this method is limited to imaging single cells or small cell aggregates on a coverslip, as it does not account for optical aberrations. A more recent approach relies on sequential and repetitive SMLM acquisition of focal planes excited by a tilted light-sheet combined with interleaved[10] or real time[21] imaging of fiduciary markers located on the coverslip. It uses long-axial range PSFs, which facilitate adjacent plane stitching but at the expenses of lower single molecule density regime due to their larger size. Additionally, its geometry restricts the achievable imaging depth to less than 15 µm above the coverslip[21,33].

In this context, we developed a solution to perform volumetric SMLM, called soSMARt for a single-objective Single Molecule Active Registration technique, which addresses the concerns of in-depth optical sectioning, depth-induced aberrations, drift correction, and 3D planes registration (Fig. 1). It is based on the single objective light-sheet microscopy technique soSPIM[19], and leverages dedicated SMARt microfabricated devices featuring 45° mirrors along micro-wells containing the samples and embedding fiduciary markers surrounding the micro-wells at all depths. These fiduciaries act as photostable point sources emitters enabling to (i) correct for in-depth induced aberrations using adaptive optics (AO), (ii) compensate for mechanical drifts in 3D during the acquisition, and (iii) allow for the precise 3D registration of the reconstructed volumes. We combined the SMARt devices with a dedicated software, called SMARtrack, that corrects for 3D drifts in real-time through an active feed-back-loop, and handles automatically the sequential multi-plane acquisition to image an entire volume. Combining the optical sectioning of soSPIM with AO and DNA-PAINT[34], we demonstrated the possibility to perform photobleaching-free astigmatism-based 3D SMLM over the entire volume of single cells ($\approx 20 \times 20 \times 10$ µm³) with a resolution (FWHM) down to $7.0 \pm 0.4$ nm laterally and $40.5 \pm 1.5$ nm axially (mean ± s.e.m., $n = 108$). While the whole acquisition can take several hours using conventional DNA-PAINT linkers and acquisition protocol, we also explored the possibility to perform deep-learning-based high density single-molecule localization[35] to drastically reduce the acquisition time by one order of magnitude. Finally, we illustrated the capacity of our method to perform 3D SMLM on different biological samples, ranging from isolated cells to more complex samples such as spheroids. We provide all the methods, including the fabrication protocol of the SMARt devices that can be used more widely than in the soSPIM configuration as reference point sources for 3D registration, calibration, and optical aberration correction for other fluorescence microscopy techniques.

## Results
### Depth-induced optical aberration characterization and correction
In-depth SMLM is impaired by optical aberrations induced both by the optical system and the sample. These aberrations distort the PSF, leading to a loss in localization precision and localization accuracy[27]. In particular, Refractive Index (RI) mismatch between the objective immersion media and the imaging media results in optical aberrations, mostly spherical, that increase with the imaging depth. Index matching the sample and the objective immersion media is thus recommended to reduce the aberrations, e.g., use Water Immersion (WI) objectives with live sample or sample in PBS solution. Yet, some RI mismatches remain, with other additional types of aberrations coming from the microscope itself.

We characterized depth-dependent aberrations on hundreds of PSF located at different depth, from 0 to 40 µm above the surface coverslip, within a phantom sample, using a WI objective. We embedded 0.1 µm fluorescent fiduciary markers within a low melting point Agarose gel, whose RI matches the imaging buffer, to act as single fluorescent emitters ("Methods"). Using a phase retrieval

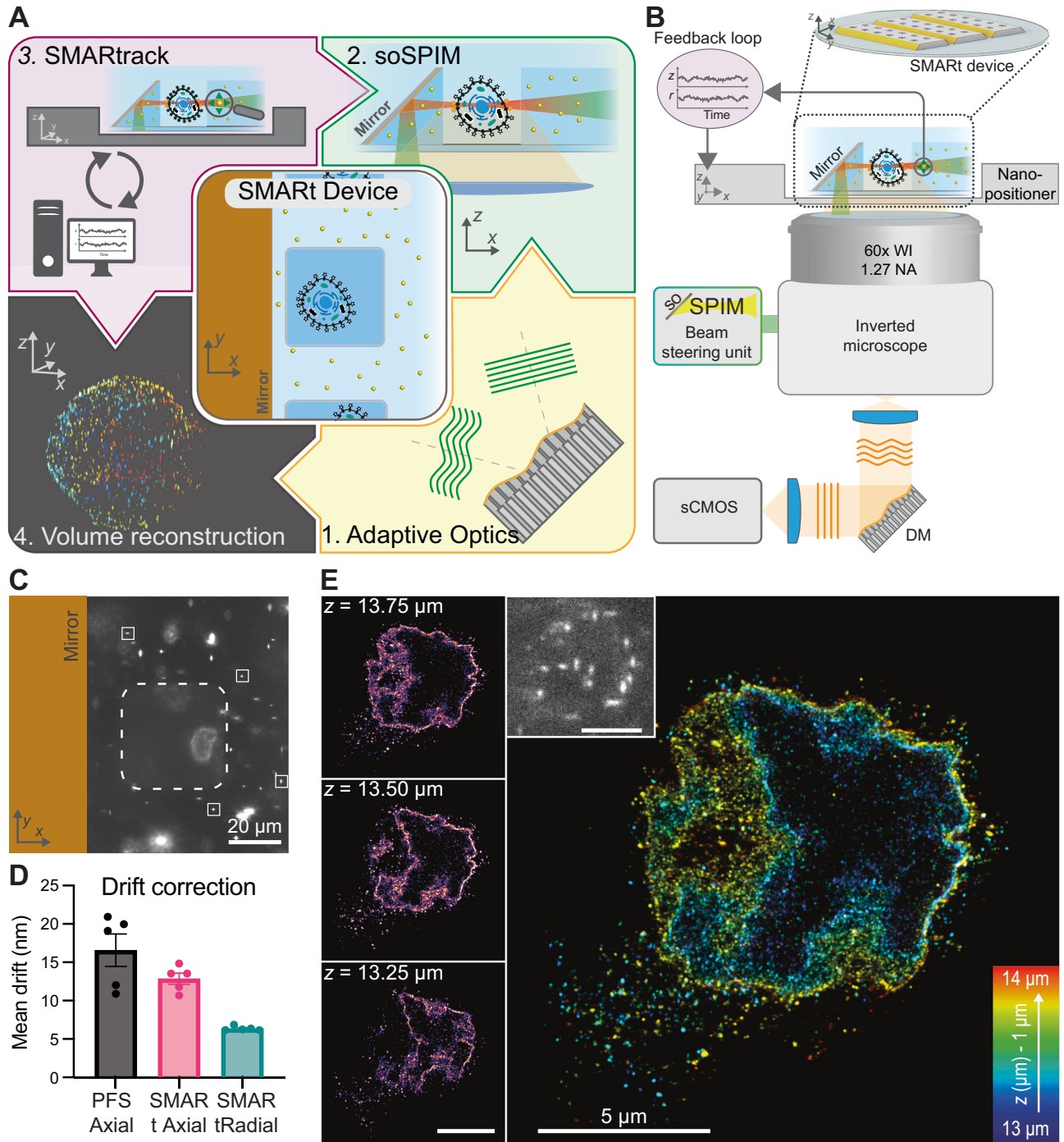

**Fig. 1 | soSMARt acquisition pipeline. A** Schematic of the soSMARt acquisition pipeline, leveraging on dedicated SMARt microfabricated devices containing 45° mirrors and embedded fiducial markers surrounding the sample. It allows (1) in-depth optical aberration correction using Adaptive Optics, (2) PSF calibration for soSPIM-based 3D single-molecule localization, (3) real-time mechanical drift correction in 3D using SMARtrack, and (4) precise extended-volume reconstruction. **B** Optical setup composed of an inverted microscope, the soSPIM beam steering unit, a deformable mirror (DM) for aberration correction, and a *XYZ* piezo stage for mechanical drift correction. **C** Image plane of the lamin B1 protein in a COS-7 cell suspended in a SMARt device well. The well containing the cell (dashed line) is surrounded by numerous fluorescent fiducial markers (four examples shown in white boxes). **D** Mean residual drift after real-time feedback-loop registration using SMARtrack, computed from the localization precision of $n = 5$ fiducial markers in the polymer of a SMARt device (mean ± s.e.m). **E** In-depth astigmatism-based 3D SMLM of the Lamin B1 nuclear envelope of a COS-7 cell reconstructed from a single plane DNA-PAINT acquisition at 13.5 μm above the coverslip. Colors indicate the *z* distance from the coverslip. Left: SMLM reconstructions at three intermediate depths. Inset shows an example of an astigmatism-based DNA-PAINT image (scale bars are 5 μm).

approach[36], we assessed the amount of aberrations present on each PSF, as a function of depth (Supplementary Fig. 2A). We first observed the presence of a stable amount of vertical astigmatism, as induced intentionally to perform astigmatism-based 3D localization. We also observed that most of the aberrations were stable with depth, except for the third and fifth order of spherical aberrations, that increase, which is consistent with the presence of a RI mismatch. The analysis of the evolution of the third order spherical aberration with depth,

showed a linear relation with a slope of $0.031 \pm 0.003$ rad.µm$^{-1}$ (Supplementary Fig. 2B).

We then characterized the effect of depth-dependent aberrations on astigmatism-based SMLM[25]. Indeed, several studies reported that depth-dependent aberrations, mainly spherical aberrations, tend to degrade the astigmatism signal, and therefore impair the capacity to accurately localize single molecules axially[28,37–39]. We induced 60 nm rms astigmatism, reported to be optimal[28], and computed the slope of the curve $(\sigma_x - \sigma_y)$ as a function of the imaging depth, with $\sigma_x$ and $\sigma_y$ being the PSF width along the x- and y-axis, respectively (Supplementary Fig. 2D). This slope relates the precision with which a molecule can be localized axially. Without correction of depth-dependent aberrations, we could observe a clear decrease of the slope with depth (Supplementary Fig. 2E). After correcting the first order aberrations using a 3 N algorithm approach based on a maximum intensity metrics computed from the image of a fiduciary embedded in the gel ("Methods"), we could partially restore the slope at depths up to 45 µm (Supplementary Fig. 2E), which is consistent with previous studies[28,37–39].

For in-depth 3D-SMLM imaging, specific depth-dependent optical aberration characterization and correction were performed using a 3 N algorithm based on the maximum intensity metric measured on stationary fluorescent beads. This was possible thanks to the dedicated SMARt micro-fabricated device embedding 0.1 µm fluorescent nanodiamonds homogeneously distributed in the RI matched polymer surrounding the microwells. Astigmatism-based 3D SMLM was then performed by inducing 60 nm rms astigmatism post-aberration correction. In addition to provide reference point sources for aberration correction, fiduciaries were also used to calibrate the astigmatism-based PSF for optimal in-depth 3D localization, and for real-time drift correction.

## Real time 3D nanoscale drift correction

As mentioned above, drift correction is critical in SMLM. In most SMLM configurations, axial drift needs to be corrected in real time, while lateral drift can be corrected post-acquisition. However, the specific architecture of the soSPIM technology, which relies on the reflection of a laser beam onto a 45° mirror integrated into the imaging device to create the excitation light-sheet (Fig. 1), necessitates both lateral and axial drifts to be corrected in real-time. Indeed, the axial position of the excitation plane, which needs to overlap with the imaging plane of the objective, is adjusted by translating the laser beam in the lateral direction, along the direction of the light-sheet propagation. As a consequence, any lateral drift in the direction of the light-sheet propagation leads to equivalent shifts of the light-sheet in the axial direction, and therefore a progressive misalignment of the excitation and the imaging planes (Supplementary Fig. 3A).

To maintain the excitation and the imaging planes superposed for long acquisition time, we developed a real-time 3D nanoscale drift correction approach using feedback loop. It leverages on two dedicated elements: First, the SMARt microfabricated soSPIM devices, in which fluorescent nanodiamond fiduciary markers have been homogeneously embedded into the RI matched bio-compatible polymer surrounding the microwells hosting the samples at a density of $(4.5 \pm 3.1)*10^{-3}$ beads per µm$^3$ (mean $\pm$ s.e.m., $n = 4$) (Fig. 1B, C and Supplementary Fig. 4). We choose nanodiamonds for their photostability, which enables them to withstand both the SMARt devices' fabrication process and extended acquisition times. Additionally, their lower brightness compared to conventional fluorescent beads makes them less susceptible to saturation under SMLM illumination conditions, thereby making them more suitable for registration. Second, a real-time drift correction software, called SMARtrack, which localizes and tracks fiduciary markers in real-time, and corrects the drift using an active feed-back loop on a XYZ piezo stage with a nanoscale precision (Fig. 1A and "Methods"). We characterized the precision of our

SMARtrack drift correction solution by measuring the mean localization precision of several fiduciary markers embedded in the device polymer to be $12.9 \pm 1.6$ nm axially and $6.4 \pm 0.3$ nm laterally (mean $\pm$ s.e.m., $n = 5$) (Fig. 1D and Supplementary Fig. 3B, C). As a comparison, the axial correction achieved by the Nikon perfect focus system (PFS) was measured to be $16.6 \pm 4.7$ nm, consistent with its specifications. However, unlike the Nikon PFS, which only compensates for axial drifts, the soSMARt configuration requires active compensation in both axial and lateral directions. The SMARtrack 3-axis drift correction system ensures that the excitation plane remains superposed to the detection plane throughout the acquisition process (Supplementary Fig. 3A). While improved performances could be achieved, e.g., by temporal averaging bead localization coordinates, this is not necessary since additional offline registration is performed during the reconstruction process using several surrounding fiduciary markers, guaranteeing optimal drift correction ("Methods").

SMARtrack also allows to select automatically the fiduciary candidates for registration amongst the different beads that are present at each plane. It is based on filtering the Gaussian fitting parameters $(\sigma_x, \sigma_y, I, \chi^2)$, aiming to select a bead that is axially closest to the imaging plane with the best localization accuracy, and filtering out any potential aggregate or dim fiduciary.

## In-depth 3D-SMLM

As a first demonstration of 3D SMLM imaging at several microns above the coverslips using soSMARt, we imaged the nuclear envelope of isolated COS7 cells with DNA-PAINT imaging[34]. COS-7 cells were seeded into a soSPIM SMARt device containing several hundreds of $40 \times 40 \times 45$ µm$^3$ micro-wells flanked with 45° micro-mirrors. Cells were then fixed and immunolabelled directly into the soSPIM imaging device for the Lamin B1 protein, using secondary antibodies functionalized with DNA docking strands before being imaged with the complementary imager strands.

Before starting the acquisition, we calibrated the system by selecting one isolated fluorescent nanodiamonds located at the imaging depth corresponding to the imaging plane, 14 µm above the coverslip (Fig. 1C). Optical aberrations were then corrected automatically using a 3 N algorithm on the 9 first Zernike modes, based on the maximum intensity metric ("Methods"), and 60 nm rms of astigmatism were induced with our DM for 3D localization (Supplementary Fig. 2D). An acquisition sequence of 30,000 frames was acquired at 10 Hz, for a total time of ≈ 50 min, without noticeable drift thanks to the SMARtrack active feedback-loop drift correction (Fig. 1C). Figure 1D and Supplementary Movie 1 illustrate the quality of in-depth astigmatism-shaped PSFs and the low background signal thanks to the combination of light-sheet illumination and AO. We could reconstruct a 1000 nm thick optical slice of the lamin B1 nuclear envelope from 297,736 localizations, with a resolution of 31 nm radially estimated by Fourier Ring Correlation (FRC)[40]. Thickness of the Lamin nuclear envelope was estimated to be $79 \pm 2$ nm radially and $183 \pm 2$ nm axially (FWHM, mean $\pm$ s.e.m.) (Fig. 1E and Supplementary Fig. 5), reflecting the convolution of the laminar envelope thickness with the radial and axial resolutions, respectively.

To demonstrate the improvement in reconstruction quality enabled by correcting depth-dependent aberrations, we acquired 12,000 single molecule frames at 30 µm above the coverslips, imaging the lamin B1 nuclear envelope. This was performed using: (1) a cylindrical lens in the detection path for astigmatism-based localization, and (2) a deformable mirror (DM) to both correct optical aberrations and induce controlled astigmatism (Supplementary Fig. 6). We found that depth-dependent aberrations significantly degrade the PSF, leading to reduced reconstruction accuracy and the appearance of artifacts, regardless of whether astigmatism was calibrated at the coverslips or at the imaging depth (Supplementary Fig. 6A, B). In particular, we observed reconstruction echoes, artifacts characteristic

of spherical aberrations when using Gaussian fitting-based localization. Correcting these aberrations with the DM allowed us to restore the PSF integrity, and achieve improved 3D reconstruction quality (Supplementary Fig. 6C). These results also confirm that the water-index-matched polymer used in the SMARt devices enables effective aberration corrections when performed near the well of interest, directly at the imaging depth[41].

## Automatic volumetric 3D-SMLM

We then demonstrated the capacity of soSMARt to perform the automatic acquisition of an entire 10 μm thick cell. The combination of light-sheet illumination with DNA-PAINT[34] allowed the sequential plane by plane acquisition with reduced background, while circumventing the bleaching issue of the fluorophores. Leveraging on the SMARt devices embedding nanodiamonds, we were able to correct for optical aberrations at the cell depth, allowing accurate 3D single molecule localization. Combined with the SMARtrack active feed-back drift correction, it was possible to acquire long-term drift-free DNA-PAINT sequences. Finally, the automatic beads selection at each plane allowed to automatize the volumetric acquisition (Supplementary Fig. 7) and reconstruction using overlapping fiduciaries.

In more details, in-depth optical aberrations were corrected using AO, and 60 nm rms astigmatism were induced for 3D localization, allowing to get consistent PSF deformation and axial resolution at any depth over the entire cell of interest. The plane-by-plane sequential acquisition process was then fully handled by the SMARtrack software as follow: at each plane, a bead was automatically selected and used as fiduciary marker for real time feedback-loop drift correction. At the end of each plane acquisition, both the image plane and the excitation plane were adjusted to the next z-position, and a new drift-free acquisition sequence was launched until all the planes were acquired (Supplementary Fig. 7). For the entire volume reconstruction, lateral drifts between adjacent planes were performed by localizing several overlapping fiduciaries, and correcting the localizations with their mean 2D shifts ("Methods").

As a demonstration of the soSMARt stability and automation, we first acquired the mitochondria network of a COS-7 cell suspended into one well of a SMARt device, tens of microns above the coverslips over a thickness of 3 μm (Fig. 2A). We acquired 6 planes every 400 nm with 8000 frames per plane at 6.66 Hz acquisition frame rate, from which 162,726 localizations were kept post-filtering for the reconstruction. The entire acquisition process took about 3 h. The spatial resolution of the reconstructed image was estimated to $55 \pm 3$ nm radially (mean ± s.e.m.) by FRC computed on 14 planes reconstructed every 200 nm. It was estimated to $11.4 \pm 0.5$ nm laterally and $49.3 \pm 2.8$ nm axially by localizing fiduciary markers embedded into the SMARt device polymer (FWHM, mean ± s.e.m., $n = 110$) (Supplementary Fig. 8A) ("Methods"). We next acquired the entire volume of the Lamin B1 nuclear envelope of a COS-7 cell suspended into a well of a SMARt imaging device (Fig. 2B). We acquired 35 planes every 350 nm, with 10,000 frames per plane at 6.66 Hz acquisition frame rate, from which 618,187 3D localizations were kept post-filtering. The entire acquisition process took about 16 h. We observed that the distribution of the number of localizations along the axial direction was rather homogeneous (coefficient of variation of 56%), illustrating the soSMARt stability of the acquisition over time and for the whole volume. We also noticed that a 300 to 400 nm plane-to-plane distance was optimal to homogeneously populate the localizations over the entire volume, as compared to higher distances (e.g., 500 nm, which shows a coefficient of variation of 65%). This is due to the lower capacity of astigmatism-based 3D localization by Gaussian fitting to localize beyond ±400 nm from the focal plane (Supplementary Fig. 9). 3D reconstructions allowed to clearly identify the full shape and slopes of the Lamin cortex around the nucleus. The lateral resolution was estimated to $78 \pm 3$ nm radially (mean ± s.e.m., $n = 45$) by FRC computation, and to

$7.0 \pm 0.4$ nm laterally and $40.5 \pm 1.5$ nm axially by localizing the fiduciaries embedded into the SMARt device polymer (FWHM, mean ± s.e.m., $n = 108$) (Supplementary Fig. 8B). We also imaged the nuclear envelope of COS7 cells cultured for 48 h into a SMARt device prior fixation, revealing the expected smooth, wrinkle-free shapes of adherent cells, in contrast to non-adherent cells (Supplementary Fig. 10). This further illustrates the live-cell compatibility of the SMARt devices, as previously demonstrated in earlier studies[19,42].

To further validate the capacity of the soSMART technique to perform quantitative super-resolution volumetric imaging, we investigated the nanoscale distribution of PD-1 and CD3 membrane receptors at the surface of suspended Jurkat T-Cells. Membrane receptors distribution is of paramount importance in many cellular processes, and the capacity to probe it at the whole cell level with nanometric spatial resolution is valuable for many studies. Using the previously described protocol, we acquired three 10 μm-thick 3D DNA-PAINT sequences, each composed of 25 to 40 planes spaced 300 to 400 nm apart. For each plane, 10,000–15,000 frames were recorded at 10 Hz. One dataset was acquired from a JPD1-labelled cell, and two from CD3-labelled cells (Supplementary Table 01). Reconstructions were generated from 154,606 (JPD1), 512,353, and 634,073 (CD3) 3D localizations (Fig. 2C and Supplementary Fig. 11). Spatial resolution, estimated by FRC, was $30 \pm 1$ nm radially for the JPD1-labelled cell, and $67 \pm 5$ nm and $63 \pm 6$ nm for the two CD3-labelled cells. Using fiduciary markers, resolution was further assessed: for PD1-labeled cells, it was $16.5 \pm 0.9$ nm laterally and $55.5 \pm 2.8$ nm axially; for the CD3-labeled cells, it was $13.5 \pm 0.4$ nm and $11.4 \pm 0.5$ nm laterally, and $56.4 \pm 1.6$ nm and $49.3 \pm 2.8$ nm axially, respectively (Supplementary Fig. 8C–E). We then performed the nanoscale quantitative analysis of PD-1 (resp. CD3) receptors at the whole cell surface using the tessellation-based analysis software Point-Cloud Analyst (PoCA)[43,44]. We quantified the cluster properties (ie. local density, lateral and axial extension, volume, number of localizations) and their spatial distribution at the Jurkat T-Cell surface by projecting the centroid of each detected cluster onto the sphere best-fitting the cell surface (Fig. 2D and Supplementary Fig. 11). Quantitative cluster properties were mapped on the Voronoi polygons computed from the positions of the surface receptors clusters on the cell surface best-fitting sphere, allowing to visually assess their 3D distribution, which appear homogenous over the whole cell membrane. We measured PD-1 (resp. CD3) clusters mean lateral sizes (FWHM, mean ± s.e.m.) of $50.5 \pm 0.4$ nm ($n = 1,086$) (resp. $56.3 \pm 0.5$ nm ($n = 1,853$) and $64.9 \pm 0.7$ nm ($n = 1,575$)) and a mean axial size of $119 \pm 2$ nm (resp. $109 \pm 2$ nm and $128 \pm 2$ nm). We also computed the distribution of these quantitative parameters as a function of the z direction (Fig. 2D and Supplementary Fig. 11). Their homogenous distribution along the z-axis illustrates the homogeneous quality of the data over the whole acquired volume.

## Fast volumetric 3D-SMLM using high-density SMLM

To overcome the long acquisition time of 3D SMLM, which can range from several hours to days[14], we combined the recent deep-learning-based high density localization method DECODE, with our volumetric imaging method. DECODE relies on the simulation of realistic PSF models[35,45] to train a convolutional neural network. It allows predicting the 3D position of single emitters in overlapping conditions with higher performances compared to any other existing model-based approaches.

We acquired 20 planes every 500 nm of a whole cell labeled for LaminB1 with DNA-PAINT, using a higher imager concentration (2.5 nM) compared to the standard conditions (0.2–0.8 nM). With only 2000 images per plane, we could successfully reconstruct a whole cell from 1.9 M of localizations in only 1.5 h, with a spatial resolution estimated to $42 \pm 1$ nm radially (mean ± s.e.m., $n = 54$) by FRC, and to $10.0 \pm 0.2$ nm (resp. $41.2 \pm 0.9$ nm) radially (resp. axially) using the fiduciaries embedded into the SMARt device polymer

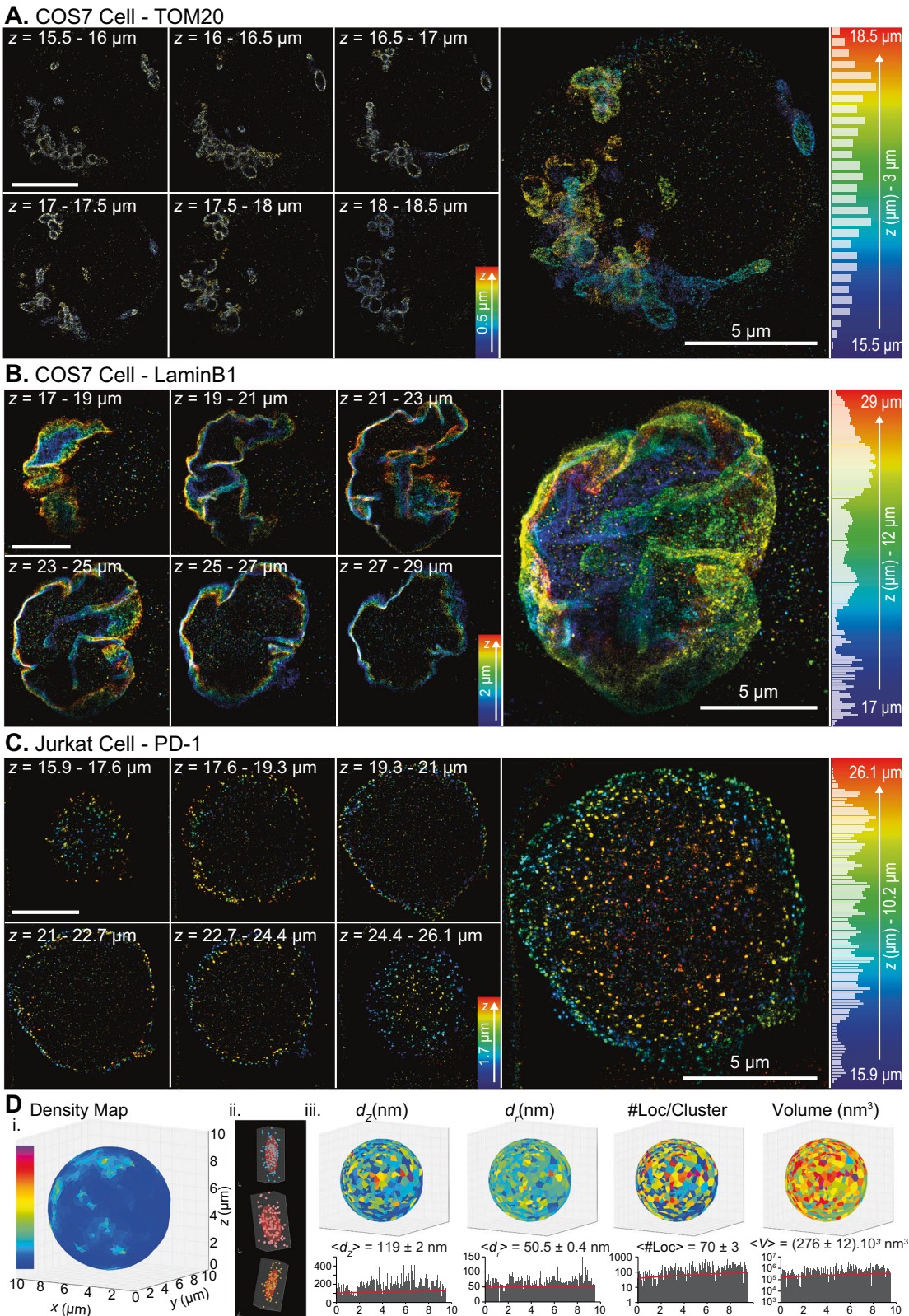

**A.** COS7 Cell - TOM20

**B.** COS7 Cell - LaminB1

**C.** Jurkat Cell - PD-1

**D** Density Map

i.

ii.

iii.

$d_z$ (nm)

$d_r$ (nm)

#Loc/Cluster

Volume (nm³)

$<d_z> = 119 \pm 2$ nm

$<d_r> = 50.5 \pm 0.4$ nm

$<\#Loc> = 70 \pm 3$

$<V> = (276 \pm 12).10^3$ nm³

(FWHM, mean ± s.e.m., $n = 299$) (Fig. 3A and Supplementary Fig. 8E). This represents a speed-up factor of about tenfold compared to standard experiments (Fig. 3B), in agreement with previously published works for whole cell single-molecule-based super-resolution imaging[21,33]. However, we could notice a systematic grid pattern overlapping with the reconstructed images, at a frequency corresponding exactly to the camera's pixel size, as also recently described by Fu et al.[45]. We suppose this artefact to come from non-homogeneous background of DNA-PAINT data, which is not correctly taken into account in the simulations used for the training data. To attenuate this grid artefact, we implemented a simple Fourier filtering, setting to zero the multiple of the frequencies corresponding to the inverse of pixel camera in the super-resolution image (Supplementary Fig. 12 and "Methods").

**Fig. 2 | Volumetric 3D-SMLM. A** 3D reconstruction of a 3 μm thick slice of the mitochondrial network of a COS-7 cell suspended in a well of a SMARt device. Left: 0.5 μm thick reconstructions at different $z$ positions. Right: 3 μm thick reconstruction with a histogram showing the number of localizations along the $z$-axis. Color indicates the $z$ distance from the coverslip. **B** 3D reconstruction of the entire Lamin B1 nuclear envelope of a COS-7 cell suspended in a well of a SMARt device. Left: 2 μm thick reconstructions at different $z$ positions. Right: 12 μm thick 3D reconstruction of the entire nucleus, with a histogram showing the number of localizations along the $z$-axis. Color indicates the $z$ distance from the coverslip. **C** 3D reconstruction of PD-1 receptors of a Jurkat T Cell suspended in a well of a SMARt device. Left: 1.7 μm thick reconstructions at different $z$ positions. Right: 10.2 μm thick reconstruction of the entire cellular membrane with a histogram showing the number of localizations along the $z$-axis. Color indicates the $z$ distance from the

coverslip. **D** (i) PD-1 cluster density map computed using the PoCA clustering software projected onto the sphere best fitting the cell membrane as defined by the centroids of the localization clusters. (ii) Three examples of PD-1 clusters with the corresponding ellipsoid fitted to the localization clusters. (iii) Top: different representations of quantitative cluster properties projected onto the sphere best fitting the cell membrane. From left to right: FWHM along the $z$-axis ($d_z$); FWHM in the lateral plane ($d_r$); number of localizations per clusters (#Loc); and volume ($V$). Bottom: Same cluster properties plotted as function of their $z$ position (black) along with the corresponding linear regression (red) ($R^2 \leq 0.03$)). The reported values correspond to the mean ± s.e.m. per cluster, for the following cluster properties: FWHM of cluster size in the axial and lateral (radial) directions, number of localizations per cluster, and cluster volume ($n = 1086$).

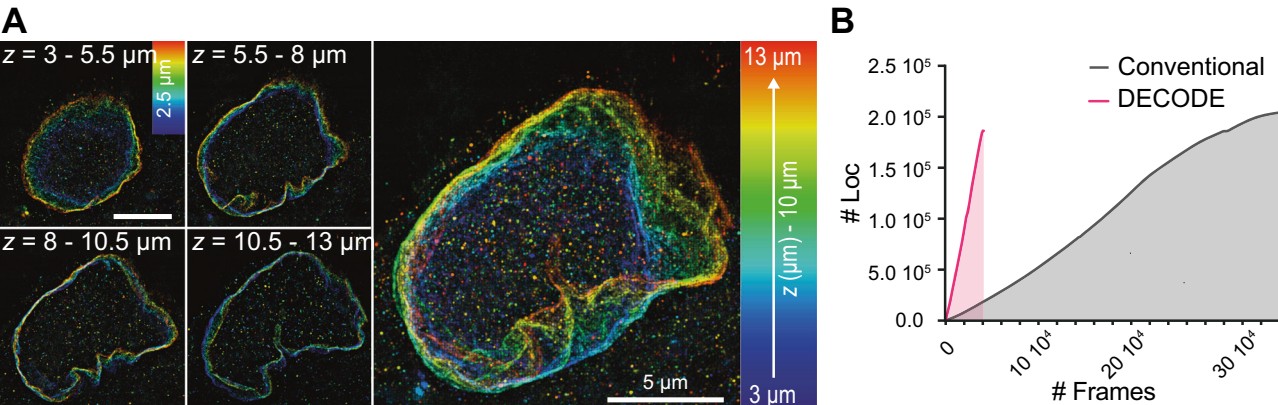

**Fig. 3 | Fast 3D-SMLM. A** 3D reconstruction of the Lamin B1 nuclear envelope of a COS-7 cell suspended in a well of a SMARt device, using DECODE deep learning-based single molecule localization and Fourier filtering. Left: 2.5 μm thick $z$ color-coded reconstructions of the DNA-PAINT acquisition at different depths. Right: $z$ color-coded reconstruction of a 10 μm-thick nucleus. **B** Comparison of the cumulative number of localizations as a function of the number of acquired frames, for high density SMLM (magenta, cell of (**A**)) and low density SMLM (grey, cell of Fig. 1B).

We can note that all the DNA-PAINT acquisitions were performed using the standard DNA-PAINT kits, which are known to have a slow binding-unbinding kinetics, limiting the achievable acquisition speed. The acquisition time could thus be further reduced using recently published strategies such as changing the imaging buffer composition[46], using fluorogenic imager strand[47], or using faster alternative docking and imager DNA sequences[48,49].

## Volumetric SMLM imaging in 3D cell cultures

Finally, leveraging the optical sectioning capability and high photon collection efficiency of the soSPIM imaging system, we conducted a proof-of-concept experiment demonstrating volumetric SMLM within 3D cell cultures.

3D cell cultures, such as spheroids or organoids, are organotypic systems that mimic real organ or tumors architecture and functions. They hold great promise for application in fundamental research, drug toxicity, and personalized medicine. However, their dimensions make their monitoring by fluorescence imaging, and super-resolution in particular, very challenging. In the case of the soSPIM, the geometry of the imaging devices, especially their size and shape, must be adapted to the biological sample under investigation. We recently introduced a new device's geometry optimized for culturing and imaging 3D cell cultures, called JeWells (Fig. 4A)[50]. However, this geometry currently does not allow for the integration of fiducial markers throughout the entire volume of the wells, complexifying sample drift and optical aberrations correction.

In this proof-of-concept experiment, we seeded HepG2 cancerous cells directly inside JeWells, which are truncated pyramidal-shaped wells with 45° reflective walls, specifically designed for 3D cell cultures and soSPIM imaging[50] (Fig. 4A). We used a JeWell device composed of an array of hundreds 100 μm height JeWells with a 70 μm top opening,

in which 3D oncospheres were cultivated for 5 days (Fig. 4B). After fixation, we labelled the nuclei envelope (Lamin B1) of the spheroids for DNA-PAINT imaging, as well as the whole nuclei with DAPI staining. We then acquired 12 planes separated by 500 nm at 30 μm above the coverslip, with 5000 frames per planes (see "Methods"), from which we could localized single molecule events and reconstruct a 3D super-resolution volume with an estimated mean spatial resolution of 54 ± 1 nm radially (mean ± s.e.m., $n = 12$, computation by FRC) (Fig. 4C). The nuclear envelope thickness was measured by Gaussian fitting of intensity profiles across the 3D volume and found to be 138 ± 13 nm (FWHM, mean ± s.e.m., $n = 20$) (Fig. 4D).

As Jewells currently do not allow the integration of fiducial markers at all depths, axial drift was corrected in real-time using the microscope's PFS, while lateral drift was corrected post-acquisition through image cross-correlation using ThunderSTORM. To further minimize the risk of lateral drift causing axial misalignment of the light-sheet with the objective's focal plane, we deliberately limited the total acquisition time.

Beyond the high photon collection efficiency and in depth optical sectioning enabled by soSPIM, the JeWell pyramidal shape also offers multi-view imaging capability. Single side illumination light-sheet illumination is often limited by sample scattering and light-sheet widening, which restrict the field of view (Fig. 4B). The ability to illuminate the sample from up to 4 sides using the JeWell geometry could significantly expand the accessible super-resolution field of view in such complex samples.

## Discussion

We introduced soSMARt, a super-resolution microscopy technique allowing to perform in-depth aberration-free SMLM. It is composed of

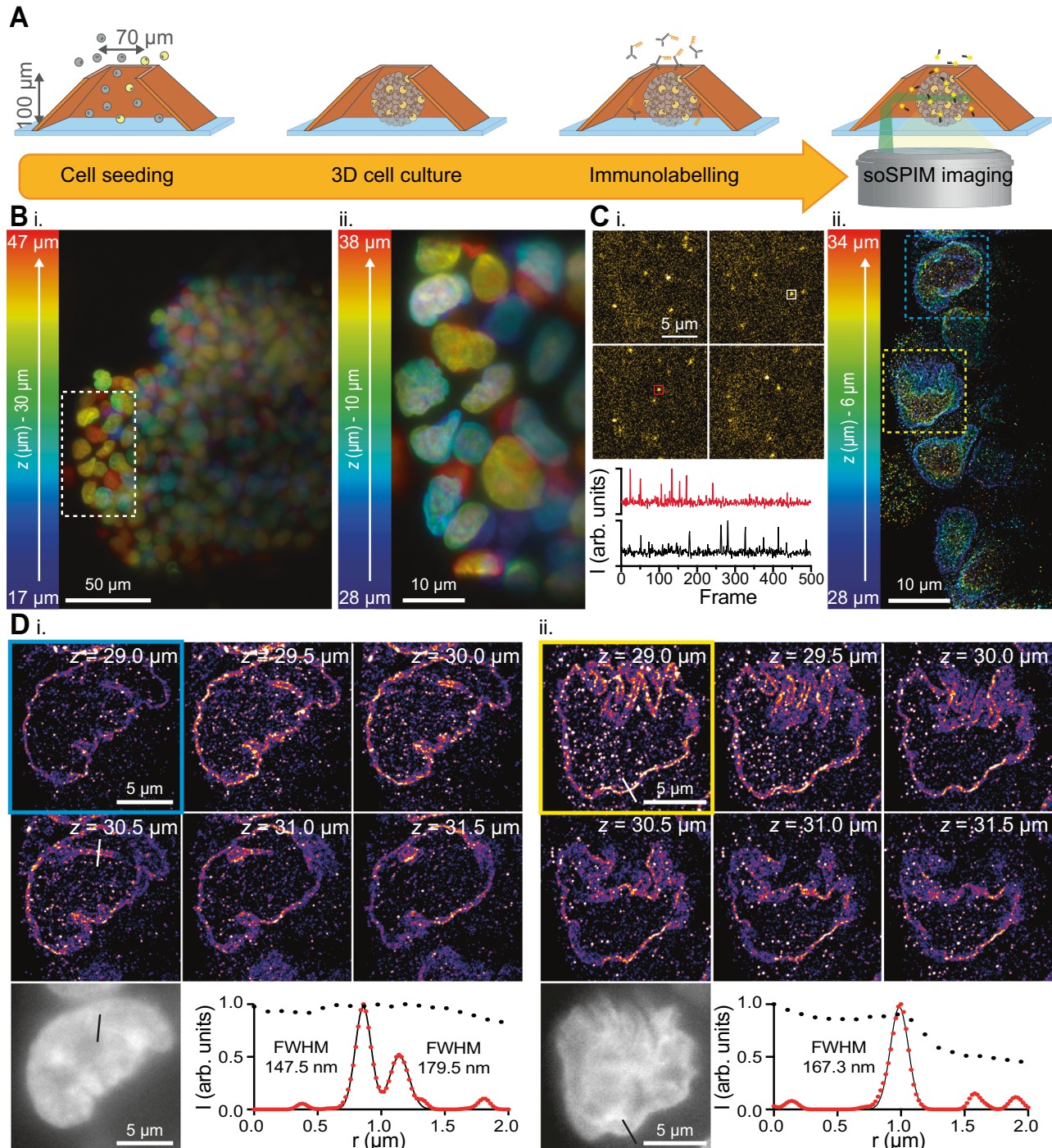

**Fig. 4 | Volumetric SMLM in 3D cell cultures. A** Schematic representation of the cell seeding, culturing, and labelling processes in the truncated pyramidal-shape JeWell used to perform soSPIM-based DNA-PAINT imaging. **B** 3D projection of a 30 μm thick volume of a Hep-G2 spheroid labelled with DAPI acquired in diffraction-limited soSPIM (i). Zoom on a 10 μm thick volume of the nucleus within the white dashed region corresponding to the volume acquired in SMLM (ii). Color indicates the z distance from the coverslip. **C** Examples of DNA-PAINT single molecule images acquired at various depths within the spheroids (top), with two illustrations of temporal intensity trace measured within the white and red regions (bottom) (i). 6 μm thick 3D DNA-PAINT reconstruction of the lamin B1 signal of the region (**B**. ii), at 28 to 34 μm from the coverslip (ii). Color indicates the z distance from the

coverslip. **D** (i) DNA-PAINT reconstructions at different depths of the cell within the blue dotted square in (**C**. ii). Bottom Left: Corresponding diffraction-limited DAPI image at 30.5 μm z distance from the coverslip. Bottom right: Line intensity profile (red dot) with the corresponding bi-Gaussian fitting (black line) along the white line in the super-resolution reconstruction. The black dots correspond to the line intensity profile on the DAPI image (black line). (ii) DNA-PAINT reconstructions at different depths of the cell within the yellow dotted square in (**C**. ii). Bottom left: Corresponding diffraction-limited DAPI image at 29 μm z-distance from the coverslip. Bottom right: Line intensity profile (red dot) with the corresponding Gaussian fitting (black line) along the white line in the super-resolution image. The black dots correspond to the line intensity profile on the DAPI image (black line).

dedicated SMARt devices, AO, and SMARtrack, a feedback-loop drift correction system that allows long term SMLM acquisition. The SMARt devices contain microwells flanked with 45° mirrors that allow for cell culture and single-objective light-sheet imaging. They embed fluorescent single emitters for real-time drift correction by active feedback-loop and 3D planes registration for entire volume reconstruction. When combined with AO, our approach enables the correction of optical aberrations at the imaging depth, while also facilitating PSF shaping and calibration for 3D single molecule localization. Notably, it allows for the direct experimental characterization of the PSF, eliminating the need for recent PSF retrieval methods[51–53], which remain complex to implement and are sensitive to variation in single molecule density and signal-to-noise-ratio. However, in its current implementation, soSMARt does not allow to correct for sample-induced aberrations. It does compensate for system- and depth-dependent aberrations, which are the predominant sources of distortion when imaging single cells or small cell aggregates[10,14,16,21,29–33]. A key advantage of the soSPIM illumination approach compared to tilted light-sheet methods is its fixed axial distance from the objective, which ensures that spherical aberrations remain uniform across the entire field of view. In the future, integrating new aberrations correction strategies, such as those leveraging on the blinking behavior of single molecules[38,53–55], could further enhance SMLM performance in more complex samples, including spheroids and organoids.

Altogether, we demonstrated that soSMARt enables automated single-molecule acquisition and reconstruction over extended volumes, with long-term nanometric sample stabilization. Our approach opens new possibilities for performing quantitative SMLM in large 3D volumes, as illustrated by several examples, including the quantitative analysis of CD3 and PD−1 membrane receptors at the surface of Jurkat T-cells, target of particular interest in immunotherapy applications. The automation of the acquisition process, combined with the unique parallelization capacities of the soSPIM devices, which can host hundreds of identical and well-aligned microwells in a single SMARt device, paves the way for a new generation of screening platforms to investigate the nanoscale effect of therapeutics antibodies. The size of the SMARt microwells can be scaled to accommodate a wide range of biological samples, from isolated cells and small cell aggregates composed of homogeneous or heterogeneous cell types, to more complex 3D cell cultures such as spheroids or organoids (Supplementary Table 1). The soSMARt platform is also compatible with live cell imaging and single particle tracking. The polymer used in the device fabrication is fully biocompatible[41,56], and previous studies have demonstrated the feasibility of using similar devices to image cells cultured for several days and adhered to the well walls[19], to visualize light-sensitive live samples[42], and to monitor the diffusion dynamics of nuclear proteins within living cells[57].

One of the most challenging, yet potentially transformative goal for SMLM, is its application to complex 3D cell cultures, such as spheroids, organoids, or tissue explants. To this end, we demonstrated the capability of soSPIM to perform 3D SMLM in such 3D cell cultures using JeWells devices[50]. However, current JeWells designs don't yet support the homogeneous embedding of fiduciary markers around the sample. Moreover, 3D cell cultures are significantly more complex than isolated single cells, and introduce additional sources of aberrations that must be accounted for SMLM. As a consequence, achieving SMLM in 3D samples with comparable quality to that of single cells will require the implementation of new aberrations correction strategies, leveraging either on the single molecule signal itself[38,53–55], or additional bulk fluorescent markers. It will also require to consider isoplanatic patch sizes to correct for spatially varying aberrations across larger fields of view, potentially requiring volume-dependent correction strategies[58]. When combined with PSF retrieval approaches[51–53] or field-dependent deep learning-based localization methods[45], we believed the ability of our platform to perform high-quality

quantitative 3D SMLM in spheroids and organoids, will have a major impact in cell biology and developmental biology in the future.

In this context, reducing acquisition time will be of paramount importance, as the inherently slow acquisition speed of conventional SMLM will remain a major limit to investigate large biological samples. A promising solution lies in deep learning-based computational approaches, which we have demonstrated can reduce the acquisition time by an order of magnitude. However, to fully harness their potential and avoid reconstruction artifacts, significant efforts must be directed toward accurate and realistic modeling of the PSF and background for convolutional neural network training.

The active feed-back loop registration implemented in soSMARt, enabled by the SMARt device, clearly demonstrates the potential of integrating smart microscopy into SMLM to enhance its performances. This concept can be extended in several promising directions, including automated cell screening, detection of rare cellular phenotypes, live identification of specific events of interest[59], and real-time analysis of single molecule data[60], coupled with automatic estimation of optical aberrations for live AO-based correction.

## Methods

### The optical system: AO-soSPIM

The soSPIM beam steering unit and AO systems were mounted onto an inverted two stages Nikon Ti2 (Microscope) equipped with a XY motorized stage (Nikon TI2-S-SE-E), an epi-fluorescence excitation path (IntensityLight C-HGFIE, Nikon) on the lower stage with GFP, CY3 and Cy5 filter sets (GFP-3035d, Cy3-4040c, and Cy5-4040c from Semrock respectively), a white light transmission path (D-LH/LC Arc lamp, Nikon), and a XYZ piezo stage (Nano-LPS200, Mad City Lab). A 60× WI Objective NA 1.27 (CFI SR Apo IR 60XC WI, Nikon) was used both for the excitation and the collection of light. After collection, the emitted light passed through the Nikon tube lens and was directed either to an Orca Flash4 V2 sCMOS camera (Hamamatsu) via a cylindrical lens for astigmatism-based PSF shaping (Manual N_STORM kit, Nikon), or into the AO module described below, prior reaching the camera. The whole microscope was enclosed in an environmental chamber (Life Imaging System) at 28 °C for improved stability purposes. The acquisition was steered by MetaMorph software (Molecular device, v.7.10.5.476).

The soSPIM beam steering unit was derived from Galland et al.[19] and coupled to the upper rear port of the microscope body (Supplementary Fig. 13). Briefly, a laser line (405 nm, 180 mW; 488 nm, 150 mW; 561 nm, 200 mW; 640 nm, 500 mW) from a laser combiner (L6Cc, Oxxius) was collimated through a 40 mm focal lens (AC254-040-A, Thorlabs) and centered into a motorized Iris (SID-1-18-L-E, SmarAct) that enables to adjust the excitation beam diameter at the objective back focal plane, and thus the light-sheet width and length at the objective image plane. The laser beam was then sent onto a focal tunable lens (Custom EL-30-10, Optotune; focal lens from −80 mm to +1,000 mm) and re-imaged onto the second mirror of a two X- and Y-axis Galvanometric Mirrors (GMs) set (Pangolin SCANMAX 506 actuators with dielectric Chroma mirrors) by two identical relay lenses (AC254-050-A, Thorlabs). A third GM (Pangolin SCANMAX 506 actuators) was inserted at the Fourier plane in between the two relay lenses and enslaved to the movement of the first (X-axis) GM to allow the virtual conjugation of the two X- and Y-axis GMs. The second (Y-axis) GM was finally conjugated to the back focal plane of the microscope objective thanks to a scan lens (Achromatic doublet 30 mm diameter 200 mm focal length) and a tube lens (Achromatic doublet 30 mm diameter 250 mm focal length). Finally, a mirror was positioned at the Fourier plane in between the scan and tube lenses in order to allow adjusting the pitch angles of the light-sheet in the objective image plane after reflection onto the 45° mirror.

The MicAO 3DSR (Imagine Optic) AO module was plugged at one output port of the microscope and steered using the MicAO software

provided by Imagine Optics. It incorporates a fifty-two actuators Deformable Mirror (DM) equipped with a temperature stabilization module (MirAO52ES) to ensure stable corrections over days. A 700 mm focal lens (14PCX700-1-1, Standa) positioned at the microscope image plane combined to the MicAOS-3DSR lenses allows to conjugate the objective back focal plane onto the DM. The objective image plane is finally imaged onto the camera after the DM. Alternatively, a Shack Hartmann WaveFront Sensor combined with a Bertrand lens to conjugate it to the DM and objectives back focal planes could be mounted instead of the sCMOS camera for the AO module calibration process.

### Depth-dependent aberrations characterization and correction

For depth-dependent aberrations characterization, 0.1 μm fluorescent microbeads at a concentration of 1:200 (Tetraspeck T7279, Thermofisher) were embedded in a drop of low melting point agarose gel (3% w/v) deposited onto a clean 1.5H glass coverslips sealed onto a bottom-free 35 mm petri dish and kept in PBS. A correction of the aberrations at the coverslips surface, using a 3N correction algorithm[61,62] as described below, was first performed to define the reference DM surface shape in order to later only characterize depth-dependent aberrations. We then acquired an epifluorescence z-stack of 50 μm from the coverslip surfaces with a z-step of 25 nm, adding 60 nm rms of pure astigmatism with the DM to characterize both the depth dependent aberrations and the astigmatism-based 3D localization efficiency as a function of the imaging depth. PSFs at various depth were then extracted and analyzed using ZOLA-3D phase retrieval plugin[36] to compute the optical aberrations as a function of depth (Supplementary Fig. 2).

The 3N correction algorithm relies on the sequential correction of a set of Zernike modes, usually the 10 first orders excluding piston, tip/tilt, and focus, by minimizing a merit factor computed from the acquired image This merit factor indirectly reflects the amount of aberration in the system. We used the inverse of the maximum intensity as a merit factor, which is well-suited for single point-source objects. The correction process involves applying positive, null and negative coefficients $(+\alpha, 0, -\alpha)$ to a Zernike mode and measuring the corresponding merit factors. A second order polynomial function is then fitted on these three points, and the coefficient α_opt that corresponds to the extremum of the function is extracted. This value represents the optimal coefficient for a given Zernike mode, effectively cancelling most of the associated optical aberrations. The process is repeated sequentially for each Zernike modes. We performed two to three iterations of this correction process on the following seven modes, using $\alpha = 60$ nm RMS: astigmatism (vertical and oblique), coma (vertical and horizontal), first-order spherical abberation, and trefoil (vertical and horizontal).

### SMARt imaging devices fabrication

Standard soSPIM imaging devices microfabrication process has been described in ref. 19 and is illustrated in Supplementary Fig. 14. Briefly, anisotropic etching in an alkaline solution was first used to create 45° slanted surface having a height of around 40 μm into a silicon wafer, followed by deep reactive ions etching to create wells of the same depth, with various XY dimensions, along those 45° surfaces (Supplementary Table 1). In this study, wells of $40 \times 40$ μm$^2$ were used. The silicon wafer features were then replicated onto 1.5H cleaned coverslips using a 2 steps process: (1) replication of the silicon wafer master mold in PDMS (Sylgard 184, Dow Corning), (2) replication of the PDMS imprint on a cleaned coverslips using a capillary filling process and UV-curing for 20 min under water (UV-Kub 2, KLOÉ) of an index-matched UV-curable polymer (BIO-133, MyPolymer).

SMARt devices are an evolution of standard soSPIM devices, designed to incorporate fluorescent fiduciary markers surrounding the wells for optical aberration correction, real-time drift compensation,

and entire volume reconstruction. We embedded fluorescent nano-diamonds into the UV-curable polymer prior to the capillary filling and UV-curing steps, as their exceptional photostability allows them to withstand the device fabrication process and very long acquisition times. Additionally, their brightness makes them particularly well-suited for registration under SMLM illumination conditions. 200 μL of fluorescent nano-diamonds from stock solution (NDNV100nmMd10ml, Adamas Technologies NC.) were centrifugated at 4500 rpm for 10 min, resuspended in 200 μL of acetone and sonicated for 30 min to prevent aggregation. Acetone solution of nano-diamonds was then mixed thoroughly within the UV-curable polymer and centrifugated at 1000 rpm for 10 min to remove air bubbles before the capillary filling process. We experimentally evaluated that a mixture of 1:20 nano-diamonds into the polymer provided optimal concentration of sparsely distributed point sources surrounding the volume of interest. We computed a nanodiamond density of $(4.5 \pm 3.1) \bullet 10^{-3}$ nanodiamond per μm$^3$, corresponding to $18.9 \pm 4.6$ beads per plane considering that each bead can be localized on 3 consecutive planes (mean ± s.e.m., $n = 4$). It was characterized by localizing all the nanodiamonds present in a maximum intensity projection of 4 z-stacks (acquisitions in Fig. 2B, C and Supplementary Fig. 11A, C) acquired around the well of interest, and dividing the number of detected nanodiamonds by the volume of the z-stack minus the volume of the well (Supplementary Fig. 4).

After curing, the PDMS mold was peeled-off, and the device was coated with a thin layer of gold by thermal evaporation in a vacuum chamber (JFC-1600 Auto Fine Coater, JEOL) to make the 45° surfaces reflective. A freshly prepared flat PDMS stamp was then deposited on top of the device and the gap created in between it and the 45° surfaces filled with a UV-curable polymer (NOA73, Norland Products) by capillary-filling. Once the polymer cured for 2 min under UV, the stamp was removed and the device was immersed in a gold etching solution (gold etchant 651818, Sigma) to remove any metal coating outside the 45° surfaces protected by the polymer layer. The coverslips were then immersed in pure Ethanol and illuminated with UV at 365 nm for two hours to bleach the polymer auto-fluorescence (UV-Kub 2, KLOE, $P = 23.3$ mW.cm$^{-2}$). Finally, the coverslip was sealed at the bottom of a bottom-free 35 mm petri-dish to allow for easy cells seeding, culturing, and labelling prior imaging.

### Cell seeding in the SMARt devices

COS7 cells (Sigma, #87021302) were cultured in high-glucose DME medium (Biowest, #L0106-500) supplemented with 10% FBS (Eurobio, #CVFSVF06-01), 1% GlutaMAX (ThermoFisher, #35050038), and 1% penicillin-streptomycin (ThermoFisher, #15140122). Immortalized human lymphocyte Jurkat cells and human PD-1 recombinant Jurkat cells (ATCC, Clone E6-1, TIB-152) were cultured in RPMI 1640-1× (with Phenol Red, GlutaMAX™ and HEPES, Thermofisher #72400047). The medium was supplemented with 10% heat inactivated FBS and 1% penicillin-streptomycin for the Jurkat cell line, plus 200 μg/mL of Hygromycin B (ThermoFisher, #10687010), 500 μg/mL of Geneticin (Gibco, #15140-122), 1 mM Sodium pyruvate, and 0.1 mM MEM non-essential amino-acids (Gibco, #11140-050) for the Jurkat PD-1 cell line

Prior cell seeding, the soSPIM imaging devices were immersed in PBS, outgazed in a vacuum chamber to remove any air bubbles trapped into the micro-wells and sterilized under UV for 15 min. PBS were then wash out and replaced by warm culture media. In the case of Jurkat cells, SMARt devices were covered with 1 mL of 5 μg/mL of Poly-L-Lysine (Sigma PLL P4707) for 45 min at 37 °C, then washed 3 times with PBS before seeding. Cells were detached by treatment with 0.2× Trypsin (Sigma), resuspended in warm culture media and 200 μL at 500,000 cells/mL were deposited on top of the micro-wells before to be placed into the incubator for 10–15 min to allow the cells to fill the micro-wells. Excess cells were then removed by rinsing with culture

media, after which 2 mL of media was added. This seeding step could be repeated two to three times, depending on the filling of the microwells by the cells. The devices were then placed into the incubators for 60 min up to 48 h for the COS7 cells, or for 2 h for the Jurkat cells, prior fixation and labelling. A short incubation time was intentionally used for COS-7 cells to collect images of partially adhered cells with wrinkled and invaginated nuclei, in order to better highlight the resolution capability of our super-resolution imaging method. Fixation and labelling steps were then performed directly into the SMARt devices as described in the supplementary methods.

### Volumetric imaging

**Imaging solution.** For DNA-PAINT imaging, the appropriate Cy3b labelled imagers ($I_1$ or $I_2$, MASSIVE Photonics) according to the docking strands used for labelling were diluted in 2 mL of the imager solution provided by the manufacturer with a final concentration ranging from 0.2 to 2.5 nM (Supplementary Table 1) and poured on top of the SMARt devices. Finally, the petri dish containing the SMARt device was sealed with parafilm and positioned onto the soSPIM microscope.

**AO correction strategy.** Depth-dependent aberrations were characterized to determine the best correction strategy.

We first measured the localization precision at 0, 10, and 25 μm above the coverslips by acquiring 1000 frames of 0.1 μm Tetraspek beads embedded in a 3% (w/v) low melting point agarose gel at each depth. Laser intensities were adjusted to acquire images with similar SNR than DNA-PAINT single molecule images and an aberration correction was performed at the coverslip surface. For 3D localization, 60 nm rms astigmatism were induced using the DM, for all depths. The pointing precision was then computed from the lateral and axial distribution of the localizations coordinates around their average coordinates. First, 3D localization for each acquired images was performed using anisotropic 2D Gaussian fitting, with a 3D calibration performed on a bead located at the coverslip. Then a Gaussian fit was performed on the lateral (resp. axial) distribution of the localization coordinates around their average coordinates, and the lateral (resp. axial) pointing precision at each depth was defined as the standard deviation of the corresponding Gaussian curves. We measured a decrease of 16 % (resp. 0%) laterally and 21 % (resp. 12%) axially of the pointing precision at 25 μm (resp. 10 μm) depth (Supplementary Fig. 2C). It illustrates the need to correct for the aberrations when imaging a cell located several tens of microns above the coverslips, but without having to update this correction over its axial extension of ≈ 10 μm.

We then characterized the slope $\left(\sigma_x - \sigma_y\right)$ as a function of depth ($z$), with $\sigma_x$ and $\sigma_y$ being the width along the $x$- and $y$-axis of the Gaussian fitting of an astigmatic PSF. Indeed, this function provides a metric of the ability to localize a single molecule along the $z$-axis in the case of astigmatism and anisotropic Gaussian fitting: the steepest is the slope, the more precise is the axial localization[28]. We acquired epifluorescence 50 μm thick $z$-stacks from the coverslip surfaces with a $z$-step of 25 nm of 0.1 μm fluorescent Tetraspeck beads embedded in a 3% (w/v) low melting point agarose gel with an aberration correction performed either at the coverslip surface, or at the beads' depth (35, 40, and 45 μm above the coverslip), and systematically adding 60 nm rms pure astigmatism with the DM. We then extracted each PSF detected according to their depth, measured their width along the $x$- and $y$-axis over an axial range of 1 μm by 2D gaussian fitting and compute the corresponding slope ($\sigma_x - \sigma_y$) as function of depth by linear regression. Finally, we pooled all the results in steps of 5 μm which correspond to 89 ± 24 beads analyzed per set of depth (mean ± s.e.m.). We observed a decrease of the slopes with the imaging depth, which is consistent with the literature[39,63] (Supplementary Fig. 2E). In particular, this loss of accuracy looks non-linear, starting from almost negligible between 0 and 10 μm depth, before

rapidly increasing until 25 μm depth. This confirms the necessity to correct for depth-dependent aberrations when imaging deeper than 10 μm. As a compromise, we therefore chose to correct for the aberrations at the middle of our imaged volume thanks to our SMARt device, and kept this correction for the acquisition of the entire volume ($\approx 10 \times 10 \times 10$ μm³).

**Feedback-loop drift correction using SMARtrack.** We used our SMARtrack MetaMorph plugin to both correct for the mechanical drifts and automatize the sequential multi-plane acquisition process as described in Supplementary Fig. 6. SMARtrack perform a feedback-loop on a $XYZ$-axis piezo stage based on the real time 3D localization and tracking of the fiduciaries embedded into the SMARt imaging device polymer.

The multi-plane acquisition process was first configured in the SMARtrack plugin, setting-up the first acquisition plane, the number of planes, the z-step between consecutive planes, and the number of frames per plane. The position of the well is identified manually by drawing a ROI, defining the region outside of the well containing the fiduciaries to localize.

During the acquisition process, one fiduciary of reference is automatically identified at each plane to perform the feedback-loop drift correction. To that end, an image is acquired before the SM acquisition sequence, from which all the fiduciaries embedded into the device polymer (i.e., outside of the well previously defined) are localized in 3D (see "reconstruction process"). A fiduciary whose axial position is close to the imaged plane (130 nm $\leq \sigma_{x,y} \leq$ 300 nm) and with a high goodness of fit ($\chi^2 > 0.65$) is then selected. When no fiduciary is identified, the acquisition process stops and is resumed to the next plane with a new fiduciary selection process. Once identified, the position of the selected fiduciary is tracked in real time during the acquisition process, and used for feedback-loop correction. The drift $d_i$ at each frame $i$ is computed as the quadratic distance between the position of the fiduciary at the beginning of the acquisition ($x_0, y_0, z_0$) and the position of the fiduciary at the frame $i$ ($x_i, y_i, z_i$) as:

$$d_i = \sqrt{\left(x_i - x_0\right)^2 + \left(y_i - y_0\right)^2, \left(z_i - z_0\right)^2} \tag{1}$$

To increase the localization precision, a user-defined temporal averaging (usually set to 10 frames) of the coordinates is performed to improve the localization precision of the selected fiduciary. Feedback-loop correction is performed frame-by frame each time $d_i$ is greater than a user defined threshold (which can be set to 0) by sending the correction to the 3-axis nano-positioner. The coordinates of all the fiduciaries used for feedback-loop drift correction at each plane are saved for later review, and used for the volume reconstruction process.

**Volumetric soSPIM-SMLM acquisition.** The camera, the microscope, the illumination devices, and the 3-axis nano-positioner stages were steered by MetaMorph software. Two home-made software (Visual Basic.net and C#), integrated as plugins into MetaMorph, allowed to control the soSPIM beam steering unit (soSPIM plugin) and to perform the automatic volumetric SMLM acquisition with the feedback-loop drift correction as described earlier (SMARtrack plugin). The *soSPIM* plugin was used to create the light-sheet by scanning a laser beam along the 45° mirror axis and to synchronize its displacement with the objective's axial movement to ensure that the excitation plane and the imaged plane remain superposed as described in Galland et al.[19].

The volumetric imaging procedure is described in Supplementary Fig. 6. After the cell of interest identification, optical aberrations at the sample depth were first corrected before inducing astigmatism for 3D localization. A fiduciary embedded into the SMARt device polymer at the depth of the cell was manually identified, and used for the 3 N optical aberrations correction algorithm as described above. Two to

**Table. 1 | Acquisition and reconstruction parameters for all experiments**

| Figure | Cell line | Labelled structure | Imager | Average imaging height (µm) | Nb planes – dz (nm) | Nb frame/plane | Frame rate (Hz) [a] | #Localization after filtering | Acquisition time |
|---|---|---|---|---|---|---|---|---|---|
| 1D | COS7 | LaminB1 | [I$_2$-Cy3b] = 0.2 nM | 14 | 1 | 30,000 | 10 | 297,736 | 50 min |
| 2 A | COS7 | TOM20 | [I$_2$-Cy3b] = 1 nM | 16 | 10 400 nm | 8000 | 6.667 | 162,729 | 3 h |
| 2B | COS7 | LaminB1 | [I$_2$-Cy3b] = 0.8 nM | 23 | 35 350 nm | 10,000 | 6.667 | 618,187 | 14 h 15 |
| 2 C | Jurkat | JPD1 | [I$_2$-Cy3b] = 1 nM | 21 | 20 500 nm | 10,000 | 10 | 154,606 | 13.2 h |
| 3 A | COS7 | LaminB1 | [I$_2$-Cy3b] = 2.5 nM | 8 | 20 500 nm | 2000 | 6.667 | 1,862,569 | 1.5 h |
| 3B. | HepG2 | LaminB1 | [I$_2$-Cy3b] 0.5 nM | 18 | 13 500 nm | 5000 | 6.667 | 931,129 | |
| Supplementary Fig. 9 | COS7 | LaminB1 | [I$_2$-Cy3b] 0.2 nM | 17 | 20 500 nm | 10,000 | 6.667 | 1,275,017 | 9 h |
| Supplementary Fig. 10 | COS7 | LaminB1 | [I$_2$-Atto650] 0.5 nM | 21 | 50 250 nm | 5000 | 25 | 6,942,324 | 3 h |
| Supplementary Fig. 11A | Jurkat | CD3 | [I$_2$-Cy3b] = 1 nM | 27.5 | 46 300 nm | 15,000 | 10 | 512,353 | 16.5 h |
| Supplementary Fig. 11B | Jurkat | CD3 | [I$_2$-Cy3b] = 1 nM | 19 | 43 300 nm | 15,000 | 6.667 | 634,073 | 14.5 h |

[a] All acquisitions have been performed using the "High-sensitivity/Slow scan" detection mode of the sCMOS camera.

three iterations of 3 N corrections on the first 7 aberration modes were sufficient to correct for all the aberrations. The quality of the correction was assessed by inducing 60 nm rms astigmatism and acquiring the 3D PSF using the selected fiduciary as point source. Calibration curves of the PSF width ($\sigma_x$) and height ($\sigma_y$) as a function of $z$ was performed and quality checked based on their symmetry. If the quality was assessed not optimal, new rounds of 3 N corrections were performed, with the possibility to correct for higher aberrations modes. The whole correction process was repeated until the correction was considered optimal. Once the calibration validated, it is used for real-time feedback-loop drift correction (SMARtrack) and volume reconstruction.

All the parameters used in SMARtrack (i.e., Imager concentration, acquisition parameters, etc.) for each acquisition are described in the Table 1.

**Whole cell reconstruction**
The reconstruction of the final super-resolution volume was performed in three steps:

Step 1 - single molecule localization: first, all single molecules were localized in 3D using PALMTracer[64] software, combining wavelet decomposition with anisotropic Gaussian fitting[60]. The remaining residual drift, due to the 3-axis nano-positioner precision and drift correction frequency, was then corrected offline using the fiduciaries selected by SMARtrack during the acquisition process. Localizations were finally filtered based on quality metrics (i.e., goodness of fit $\chi^2 \in [0.6;1]$ and axial localization range: $z \in [-0.5;0.5]$) in order to keep the localizations with the best precision for the super-resolution reconstruction.

Step 2 – plane to plane registration: for each plane $i$, multiple (4 to 10) localized fiduciaries present on both plane $i$ and adjacent plane $i+1$ were manually selected, and their lateral shifts between the two planes were measured. The single molecule coordinates of plane $i+1$ were then corrected laterally by the average of the previously measured shifts.

Step3 – volumetric reconstruction: For each plane $i$, a vertical offset of $(i-1)^*\triangle z$ was added to all localizations, where $\triangle z$ corresponds to the $z$-step of the 3-axis nano-positioner.

The homogeneity of the localization distribution along the axial direction was evaluated by computing its coefficient of variation, defined as the standard deviation of the localization distribution divided by its mean.

The reconstructed volumes were rendered using the ThunderSTORM ImageJ plugin[65], with a pixel size of 10 × 10 nm², a plane spacing of 200 nm, and a fixed Gaussian blur of 10 nm (resp. 50 nm) in the lateral (resp. axial) directions. For illustration purpose, localizations depth was color coded using the ImageJ Hyperstack/Temporal-Color code.

**Cluster analysis**
PD-1 and CD3 cluster analysis were performed using Point Cloud Analysis software[66]. First, the 3D Voronoi diagram was computed from all the localizations after the volumetric reconstruction process described above, in order to extract the local localization density[44]. Clusters were defined as aggregates containing at least 10 localizations, with a local density greater than the average density, and a maximum localization distance (cutting distance) of 50 nm.

For PD-1 expressing cells, we computed the sphere that best intercepts the centroids of all localization clusters. The cluster centroids were then projected on the sphere surface, and a 2D Voronoi diagram was generated along the sphere. This enabled to compute the local cluster density on the approximated cell membrane, providing insight to the spatial distribution of the clusters across the cell surface.

**Spatial resolution estimation**
The spatial resolutions of the reconstructed images were estimated using two different methods:

As a first method, clusters of localizations from beads embedded in the SMARt device polymer were computed and analyzed using Point Cloud Analysis software[66] as described above. Close beads were separated using a second level clustering as describe in ref. 44 using a local density threshold equal to the average local density. Then, all the localization clusters were fitted using a 3D ellipsoid model, and projected along the three main axis of the ellipsoid. Finally, the lateral and axial resolutions were computed by 2.35 times the standard deviation of the localization projections along the short and the long axis of the ellipsoid, respectively, equivalent to the FWHM of a Gaussian distribution.

As a second approach, we used the FRC[40]. Two 3D intensity stacks of the entire volume were reconstructed using the ImageJ ThunderSTORM plugin[65] from the even and odd localization identifiers, with a

pixel size of $10 \times 10 \text{ nm}^2$, a plane distance of 200 nm, and a fixed Gaussian blur of 10 nm in the lateral direction and 50 nm in the axial direction. The FRC spatial resolution was computed for each plane of the reconstructed stack using the ImageJ plugin BIOP/FRC, with a resolution criterion of 1/7. The overall FRC spatial resolution of the 3D stack was then determined by calculating the mean ± s.e.m. of all the FRC values across all reconstructed volumes.

### In-depth SMLM imaging of 3D cell cultures

**Cell seeding and culture in JeWells.** JeWells with a top opening of 70 μm and a height of 100 μm were fabricated according to the protocol described in refs. 50,67 and prepared for cell seeding and culturing as described in the supplementary methods. For cell seeding and culturing into the JeWells, Hep-G2 cells stably expressing H2B-GFP fusion protein were maintained in DMEM media (11965092, Invitrogen) supplemented with 10% FBS (10082147, Invitrogen), 1% GlutaMAX (35050061, Invitrogen), 1% penicillin-streptomycin (15070063, Invitrogen), and 1% Sodium Pyruvate (11360070, Invitrogen), referred as complete DME media hereafter, at 37 °C and 5% $CO_2$. After being trypsinized, the cells were suspended in complete DMEM media, centrifuged at 1000 rpm for 5 min and the cell suspension was adjusted to $0.5 \times 10^6$ cells/ml. 200 μL of the cell suspension was then poured onto the JeWell plate for 10 min to allow for the cells to fall within the pyramidal shaped wells and get approximately 50 cells per JeWell. Excess cells were removed by gently washing the plate with complete DMEM. After the wash, 2 ml of complete DMEM media was added to the device and the cells were cultured for 5 days at 37 °C and 5% $CO_2$ allowing them to form spheroids of approximatively 100 μm of diameter (Fig. 4A). Fixation and labelling steps were then performed directly into the JeWell devices as described in the supplementary methods.

**3D cell culture DNA-PAINT imaging.** Cy3b $I_2$-imager were diluted in the manufacturer imaging solution at 0.5–2 nM (Table 1) and poured onto the JeWells. The JeWell devices were then positioned onto the soSPIM microscope sample holder and imaged directly onto an ORCA Flash4 V2 sCMOS camera (Hamamatsu) without going through the AO module. The acquisition was steered by MetaMorph software and the axial drift was corrected in real time using the PFS of the microscope (Nikon) during the acquisition. Each volume was constituted of tens of planes spaced by 500 nm, with thousands of frames per plane (Table 1).

### High-density single molecule localization

High-density single molecule localization was performed using the deep learning-based DECODE software[35] available on https://github.com/TuragaLab/DECODE. It relies on three main steps: (i) the acquisition of experimental PSFs of the microscope, and the extraction of the corresponding analytical model using spline coefficients; (ii) the training of the network through the simulation of realistic ground-truth high-density datasets using the analytical PSF model determined earlier; and (iii) the prediction of the single molecule localizations from the experimental data-set using the trained model.

To train the convolutional neural network (CNN), we first acquired 45 PSFs placed throughout the field of view corresponding to a well hosting the cell to image. After the selection of the cell, we corrected the optical aberrations as described above, and induced 60 nm of astigmatism using the DM. We then moved the stage to a region containing 3 beads within the FOV, which we translated to cover the entire imaging area. At each position, a soSPIM z-stack was acquired with a 10 nm step size over a 3 μm axial range. The resulting set of 45 acquired 3D PSFs was then imported into the SMAP software to extract the spline coefficients of the analytical PSF model.

Once the analytical PSF computed, we trained a DECODE model to localize single molecules in high density, astigmatism-shaped PSF datasets. The training was performed using low-density soSMARt 3D single molecule datasets, along with the following experimental parameters from our imaging platform: signal intensity in the range of 3500 ± 1000 A/D counts per localization, background values between 0 and 200 A/D counts, a camera baseline of 100 A/D counts, a conversion factor of 0.46 photoelectrons per A/D counts and a quantum efficiency of 0.82. Training was performed on a Windows 10 64-bit operating system equipped with a NVIDIA GeForce RTX 3090 GPU and required approximately 4 h for 300 epochs. The model convergence was characterized by a Jaccard index of 0.864. The resulting model was then used plane by plane to predict the 3D coordinates of single molecules from the experimental high-density dataset.

For the acquisition, we recorded 20 planes of 2000 frames, with a plane distance of 500 nm. The imager strand concentration was set to 2.5 nM, which is about 5 times higher than the standard DNA-PAINT concentration used to image the Lamin nuclear envelope. The final volume reconstruction was performed as previously described, using the localization predictions generated by DECODE.

Grid artifacts visible in the reconstruction generated from DECODE predictions were removed using a custom ImageJ macro. It applies a Fourier filtering, by setting to 0 the pixels located at the multiples of the frequency $f = \pm 1/(\Delta^* z_{sr})$ along both axes of the Fourier Transform, with $\triangle$ the camera pixel size and $z_{sr}$ the zoom factor of the reconstructed image (Supplementary Fig. 10).

### Reporting summary

Further information on research design is available in the Nature Portfolio Reporting Summary linked to this article.

## Data availability

The single-molecule localization data generated and analyzed in this study, as well as the source data used to produce the graphs, have been deposited in the Zenodo database under accession code [https://zenodo.org/records/15168650][68].

## Code availability

The source code for the MetaMorph plugin software enabling the control and the synchronization of the soSPIM beam steering unit with the acquisition process is available on the GitHub repository: https://github.com/jbsiba/soSPIM. The source code for the SMARTrack feedback loop software is available on the GitHub repository: https://github.com/jbsiba/ZTrack. The source code for the SR_FFT_Filtering ImageJ texture filtering macros is available on the GitHub repository: https://github.com/jbsiba/SR_FFT_Filtering. The source code for the PoCA SMLM analysis software is available on the GitHub repository: https://github.com/flevet/PoCA

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

## Acknowledgements

We acknowledge F. Saltel for his kind gift of the Hep-G2::H2B-GFP stable fluorescent cell line, as well as Imagine Optic and Nikon Instrument France for their support. This work was supported by the Ministère de l'Enseignement Supérieur et de la Recherche (ANR-10-INBS-04 FranceBioImaging, ANR-21-CE44-0019 PolyFADO R.G., ANR-20-CE11-0013 STABLE-FP J.-B.S., ANR-21-CE11-0031 BAC-MMEP J.-B.S.). The authors acknowledge funding from the ANRT (CIFRE n°2017/0901 M.C. and n°2021/0866 I.I.). We would also like to thank the IINS's Cell Biology Facility and logistic staff, as well as the Bordeaux Imaging Center.

## Author contributions

R.G., H.F., and C.B. developed and constructed the imaging platform. R.G. and M.E.S.-L. fabricated the SMARt imaging devices from master molds designed and fabricated by V.V. and G.G. M.C. developed the SMARtrack software using the localization algorithm developed by J.-B.S. R.G., H.F., and M.C. seeded and labelled the cells into the SMARt devices. R.G., H.F., and J.R. characterized the depth-dependent aberration and performed the 3D SMLM soSMARt acquisitions. L.B. and A.N. adapted and run the DECODE algorithm according to the soSMARt imaging condition with the help of L.-R.M. and J.R. F.L. analyzed the SMLM dataset with the PoCA software. I.I. and R.G. fabricated the JeWell imaging devices, made grow the spheroids in them and L.B. performed the acquisition. J-B.S. conceived and co-developed the single-molecule localization software PALMTracer with C.B. R.G., and J.-B.S. conceived the study and supervised the research. All the authors contributed to the writing and revision of the manuscript.

## Competing interests

The authors declare no competing interest.
