## [Transparent Peer Review file · Nature Communications]

In-depth single molecule localization microscopy using adaptive optics and single objective light-sheet microscopy

Corresponding Author: Dr Remi Galland

Version 0:

Reviewer comments:

Reviewer #1

(Remarks to the Author)

The authors describe soSMART, an approach where they add embedded fiducial beads at various heights to their previously published soSPIM approach (Galland et al. 3D high- and super-resolution imaging using single-objective SPIM. *Nat Methods* 12, 641–644 (2015)) to enable real-time drift correction, adaptive optics for correction of depth-induced aberrations, and simplified stitching of multiple image slices. The acquisition pipeline is then automated for a simplified workflow.

Their approach is useful to the field and should in principle improve long-term imaging by the active drift stabilization, the achievable precision and resolution through the adaptive optics, and the ease and time required for slice stitching. Their approaches are mostly well-described and most claims are supported by their data. However, the authors should address the following points to clarify the performance of their system, compare their approach to previously published approaches, and enable reproducibility.

1. The authors should include comparisons to previous approaches for live drift correction used for single-molecule imaging at depth. Some of these are shown in SI Fig. 1, but the pros and cons of these approaches are not clearly outlined. For example, fiducial-free drift correction has been demonstrated for 3D SMLM (Wester et al. Robust, fiducial-free drift correction for super-resolution imaging. *Sci Rep* 11, 23672 (2021)). Also, separate detection of non-fluorescent beads can be used for 3D drift correction over an extended depth range (Coelho et al. Ultraprecise single-molecule localization microscopy enables in situ distance measurements in intact cells. *Sci. Adv.* 6, (2020)). This approach was recently demonstrated to work well in a similar light sheet work, while providing better lateral and axial precision (Saliba et al., Whole-cell multi-target single-molecule super-resolution imaging in 3D with microfluidics and a single-objective tilted light sheet, *Nat. Commun.*, 15 1-17 (2024)). This work also used a single-objective light sheet, active drift stabilization, and DECODE for tenfold improvement in acquisition speed for 3D whole-cell DNA-PAINT imaging, so the authors should compare their design to this work and add it to SI Fig. 1.
2. In terms of both live drift correction and slice stitching, the pros and cons of long axial range point spread functions should be discussed. These can be used to cover the whole axial range of cells, as shown in e.g. (Nehme et al. DeepSTORM3D: dense 3D localization microscopy and PSF design by deep learning. *Nat Methods* 17, 734–740 (2020)) and (Nakatani et al., Long-axial-range double-helix point spread functions for 3D volumetric super-resolution imaging, *J. Phys. Chem. B*, 128 (2024)).
3. Regarding AO, the authors should discuss their work in context with previous similar work using AO together with a single-objective light sheet (Hung et al., Adaptive optics in single objective inclined light sheet microscopy enables three-dimensional localization microscopy in adult *Drosophila* brains, *Front. Neurosci.*, 16 (2022)). Here the authors should discuss why their approach is better than the in situ calibration of PSFs from the single-molecule data itself used by Hung et al: INSPR (Xu, F. et al. Three-dimensional nanoscopy of whole cells and tissues with in situ point spread function retrieval. *Nat. Methods* 17, 531–540 (2020)), as well as other approaches such as DL-AO (Zhang, P., Ma, D., Cheng, X. et al. Deep learning-driven adaptive optics for single-molecule localization microscopy. *Nat Methods* 20, 1748–1758 (2023)), VISPR (Yang et al., "Accurate 3D single-molecule localization via vectorial in situ point spread function retrieval and aberration assessment," *Photon. Res.* 12, 2447-2461 (2024)), etc. It would also be useful if the authors could demonstrate and quantify that using nanodiamonds outside the well for AO successfully corrects the PSFs of the single-molecule data in the sample, where there will be additional sample-induced aberrations present. It would also be useful for the authors to discuss effects

of field-dependent aberrations, especially in the context of imaging large fields of view as shown in Fig. 3B.

4. The authors should discuss the relatively large localization uncertainty of their nanodiamonds used for live drift correction (29.6 nm and 14.6 nm axially and laterally, respectively), both in terms of why the uncertainty is this large compared to the typical localization precision of conventional fluorescent beads, and any measures you need to take, e.g. averaging etc, to not scramble/convolve your single-molecule data with this uncertainty in fiducial localization. The authors do briefly describe in the Methods that temporal averaging can be selected to improve the precision, but it would be useful to discuss the effect and need for this more in the main text.

5. In Fig. 3B, the authors demonstrate imaging of 3D cell cultures. However, for these measurements, the new SMART devices and their beneficial aspects (embedded fiducial beads for real-time drift correction, AO, and simplified slice stitching) were not used, so these data seem disconnected from the main story of the manuscript. Also, the authors do not demonstrate how well this imaging performs compared to the data shown in Fig. 2 in terms of precision and FRC resolution. The data presented in Fig. 3A is simply supporting previous studies showing about a tenfold speed improvement when using DECODE (and other deep-learning-based approaches) compared to fitting-based analyses, see e.g. (Speiser et al. Deep learning enables fast and dense single-molecule localization with high accuracy. *Nat Methods* 18, 1082–1090 (2021)), (Saliba et al., Whole-cell multi-target single-molecule super-resolution imaging in 3D with microfluidics and a single-objective tilted light sheet, *Nat. Commun.*, 15 1-17 (2024)), and (Nakatani et al., Long-axial-range double-helix point spread functions for 3D volumetric super-resolution imaging, *J. Phys. Chem. B*, 128 (2024)).

6. On line 86, the authors state that the performances of 3D cross-correlation approaches have not yet been demonstrated. The authors may want to rephrase his statement, as 3D cross-correlation has been used e.g. for slice stitching (Saliba et al., Whole-cell multi-target single-molecule super-resolution imaging in 3D with microfluidics and a single-objective tilted light sheet, *Nat. Commun.*, 15 1-17 (2024)).

7. On line 111-112: The authors state that they achieve resolution of 7 nm laterally and 40 nm axially. Please clarify if this is actually resolution (as from FRC calculations) or localization precision. None of their shown FRC curves show a 7 nm resolution. Please also clarify on line 241.

8. On line 137 and throughout: please match the terminology between the main text and the figures. On line 137, the authors mention third and fifth order spherical aberrations, whereas in e.g. SI Fig. 2 the authors refer to primary and secondary spherical aberrations.

9. Throughout the manuscript, please make sure to describe what errors indicate, e.g. standard deviation or SEM, and the number of measurements (n). See e.g. lines 175, 179, and 604.

10. On lines 202, 228, 241, and 273: The authors provide a single resolution estimation (31 nm, 55 nm, 78 nm, and 42 nm, respectively). Please state if this is estimated in full 3D (i.e. FSC), or which plane this refers to (e.g. xy). Importantly, the authors should provide the FRC resolution values both laterally (xy) and axially (xz and/or yz) for all data throughout the manuscript, including the data shown in Fig. 3B, to demonstrate the achievable performance of their system for various samples.

11. In the Results section, the authors mention that the data is filtered in post-processing. Please report all filters used.

12. For the lamin B1 reconstruction in Fig. 1, the authors report a very different thickness (79 nm vs 182 nm) in the lateral and axial direction. Is the lamina expected to vary that much in thickness at various parts of the cell, or is this a result of the different resolution achieved laterally and axially, convolving the measured thickness? The authors should discuss this.

13. On line 203, the authors likely refer to Fig. 1E rather than Fig. 1D.

14. For the suspended Jurkat cell imaging in the Results section (starting at line 243), please indicate at what height or height range above the coverslip these cells were imaged at.

15. On lines 260-261, the authors provide three lateral and three axial cluster sizes, but only specify two targets (PD-1 and CD3). Please clarify.

16. Lines 283-284: When discussing faster alternative docking and imager DNA sequences, the authors should also include the work by (Chung et al. Fluorogenic DNA-PAINT for faster, low-background super-resolution imaging. *Nat Methods* 19, 554–559 (2022)).

17. Please provide the used values or ranges for the DECODE training parameters (lines 764-765).

18. It would be useful to add a discussion about live-cell compatibility of their system for readers interested in using this approach for live-cell studies in the future.

19. Please add a schematic of the optical setup as a new SI Figure. The authors mention that the soSPIM beam steering unit was derived from their Galland et al. paper, but there are notable changes, such as the optics used to conjugate the objective back focal plane onto the deformable mirror for AO.

20. Also please add a schematic of the full fabrication pipeline. This is a key component of the paper, and there are differences between this fabrication pipeline and that which was published in their Galland et al. paper.

21. Please clarify the need for using nanodiamonds compared to regular fluorescent beads in the main text. The authors briefly touch upon this in the Methods, but this is an important consideration to discuss.

22. The authors mention that COS7 cells were fixed 60 minutes after seeding. Please include a motivation for this relatively short time frame, as the cells have limited time to attach during this time. This is particularly important to mention if the reason is that the chambers are not compatible for long term live-cell work.

23. In the methods, the authors mention that the index matched polymer is illuminated with UV for two hours to bleach the polymer auto-fluorescence. Please include the intensity of the light used.

24. Please include the camera settings used for all measurements.

25. Please include the rendering settings used for all reconstructions.

26. Please include the computer and GPU specifications used for the DECODE training and analysis for reproducibility.

27. Please include the sequences and/or the order number and company for all used DNA PAINT kits.

28. For all main and SI figures with 3D reconstructions, it would be useful if the authors specify if the number on the colorbars indicate the full z range, or they can add z-ticks to more clearly show the z values. In Fig. 2, please indicate the imaged z-range for each panel. SI Fig. 10 lacks the z indication.

29. SI Fig. 1: Please increase the size and font size to make it more legible.

30. SI Fig. 2: Please state what the error bars indicate. Also, on line 37, please describe what the error indicates ($n = 84 \pm 19$).
31. SI Fig. 3: Please add zoom in on a small part of the y-axis to visualize drift and jumping when correcting. Please quantify the stability for the two smart methods. From here, it's difficult to say if XY + PSF or XYZ is the best, and how they differ from just PSF. Also please include if averaging was used for correction or if frame-by-frame correction was used, for each data set shown.
32. SI Fig. 4: Please show also with adjusted contrast to show the cells, which cannot be seen in the current version. The bead densities noted here are different from the density noted in results ($4.5 \pm 3.1 \times 10^{-3}$). Please clarify or explain the range.
33. SI Fig. 8: Please add this data for all image acquisitions presented, and show the FRC calculations both laterally and axially.
34. Code availability: please make the SMARtrack and soSPIM plugin codes and the custom ImageJ macro used to reduce grid artefacts from DECODE-analyzed data available on e.g. GitHub.
35. Data availability: please make the source data presented in this manuscript available through e.g. Zenodo.

Reviewer #2

(Remarks to the Author)

Cabillic et al. report a new 3D SMLM technique that allows high SNR, volumetric super-resolution imaging of cells and organoids with extremely high stability. For that, the authors extended their previous approach—single objective selective plane illumination microscopy (soSPIM)—where microchambers with reflective surface provides thin illumination. Unlike soSPIM, a new approach introduced fluorescent nanodiamonds around the chamber which enables depth-dependent aberration correction and precise 3D drift correction. These issues are quite challenging to be solved but the authors demonstrated its feasibility with rigorous and quantitative analysis. The reviewer could clearly see the potential of this approach in the bioimaging community. However, a few major concerns were found, which needs to be clearly addressed before publication.

1. The first concern starts from laminB1 structure of COS-7 Cells in Fig. 1E and Fig. 2B. For healthy condition, laminB1 should show oval, wrinkle-free shape but all the images display nucleus wrinkling, depicting abnormal phenotype. This is likely due to the fact that COS-7 cells were not able to properly adhere to the surface and/or to communicate with other cells. Considering the physical dimension of the microfabricated device, it is unlikely that the chamber accommodates multiple COS-7 cells. In contrast, for Jurkat T-cells, the distribution of membrane proteins looks normal because they are floating cells. It means that soSMART would not be suitable for adherent cells.
2. A follow-up question is that it was hard to figure out how suspended COS-7 cells were immobilized on the chamber. Is their basal surface partially attached to the surface? Brightfield images would be helpful to confirm that.
3. The next concern is the layout of the manuscript. It has two separate stories—imaging cells by soSMART and imaging organoids by JeWell + soSPIM. I fully understand why the authors wanted to include the latter but it's technically different. It didn't exploit fiducial nanodiamonds which is the core of the manuscript, I believe.

There are also minor comments below:

1. Light sheet thickness and length need to be reported.
2. Are there any shadow effects on the excitation beam due to the embedded nanodiamonds?
3. It is crucial to show 3D DNA-PAINT imaging without and with AO correction. Supplementary Figure 2E is not sufficient to see the effect.
4. Several typos/errors were found for example:
Line 90: ref30 is incorrect.
Line 225: 400 um  400 nm
Line 443: 700 nm  700 mm
Supplementary ref1 needs to be switched with ref2.

Reviewer #3

(Remarks to the Author)

This manuscript presents an improved single molecule localization microscopy for deeper biological process investigation. The authors used microfabricated devices for light-sheet microscopy, adaptive optics for aberration correction, and real-time feedback for drift correction. The approach was applied to assess 3D nanoscale protein organization in (fixed) cells. Overall, the manuscript is well written and seems scientifically sound.

Comments:

Figs. 1-3: Regarding the color coding of the position along the z-axis: To my understanding the value in micrometers printed on the color bars indicate the ranges from the blue to the red color. I think it would help the reader to instead overlay the color bars with axes with ticks indicating a sequence of corresponding z positions, e.g., 1 um, 2 um, 3 um, etc.

Drift. The authors went to great lengths to correct for drift during the acquisition (which is a good thing). Yet, to me it is unclear what the (main) sources of drift are. Is it mechanical drift of the microscope? Z-drift induced by evaporation of the immersion water? Or sample drift due to contraction/expansion of the hydrogel, or something else? It would be good to

understand and discuss the underlying causes. For example, if the evaporation of immersion water over several hours of imaging is a problem, some immersion oils with water-like refractive index could be used.

Fig. 3B iii. This figure panels shows a typical challenge with light sheet microscopy: The cells on the right side, which are further from the incident light sheet, are poorly resolved because the light sheet is distorted and scattered passing through the cells on the left (close to the incident light sheet). Would it make sense to use self-reconstructing Bessel or other (non-Gaussian) beam shapes? Or dual illumination from both sides? I understands this complicates the experimental setup, but I think these challenges should be at least mentioned and discussed.

Fig. 3B ii&iii. Somehow the inset of ii) magnified in iii) looks different. Maybe it is because of the color code rescaling to a different range? It would be good to show the data in the same way for consistency.

Live cell usability. If I understand correctly, all data shown were acquired in fixed cells. It would be fantastic if the 3D-SMLM approach shown in this manuscript could be applied to live cell samples as well. The long acquisition times are of course challenging and the required temperature control to typically 37C will accelerate any drift. It would be good if the authors could discuss if and how this method could be applied to live samples.

Figure readability. Some of the fonts are way too small, for example the scale bar labels in Fig 3. While a PDF copy allows to zoom in, the authors should ensure to maintain readability when printed out on A4 paper.

Version 1:

Reviewer comments:

Reviewer #1

(Remarks to the Author)

Overall, the authors have made substantial revisions that have further strengthened the manuscript. They have satisfactorily addressed all but one of my previous comments.

7. The authors still refer to the localization precision of the nanodiamond data as “resolution”, and report the resulting localization precision as resolution in the same sentences as when discussing the FRC resolution of the single-molecule data (see e.g. lines 125-126, 270, 284, 301-303, and 327). This is very likely to confuse the readers. Particularly, on line 125-126, the useful metric to report is the achievable FRC resolution from the single-molecule data rather than the achievable localization precision of nanodiamonds, and this should be changed. The authors more clearly state this distinction in their SI Fig. 8, where they state “Beads localization precision” and “FRC per plane”.

I also strongly encourage the authors to make the differences between the data acquisition approaches in Fig. 3B vs the rest of the manuscript very clear (pointed out by both reviewer 1 (comment #5) and reviewer 2).

(Remarks on code availability)

Reviewer #2

(Remarks to the Author)

The authors have clearly addressed all the concerns that I had. I don't have any further comments.

(Remarks on code availability)

Reviewer #3

(Remarks to the Author)

In their revised manuscript, the authors have reasonably addressed my questions except one:

Question: The authors went to great lengths to correct for drift during the acquisition (which is a good thing). Yet, to me it is unclear what the (main) sources of drift are. Is it mechanical drift of the microscope? Z-drift induced by evaporation of the immersion water? Or sample drift due to contraction/expansion of the hydrogel, or something else? It would be good to understand and discuss the underlying causes. For example, if the evaporation of immersion water over several hours of imaging is a problem, some immersion oils with water-like refractive index could be used.

Answer in the rebuttal: Drift is a common issue in microscopy. It is particularly visible and negative in Single Molecule Localization based Super-Resolution Microscopy, due to the long acquisition times (ranging from minutes to hours) and the aim for nanometric resolutions. Main sources of drift in microscopy are likely due to thermal fluctuations causing the contraction and expansion of components such as the microscope or stage, as well as potential vibrations from the microscope and camera. While these factors usually have a minimal visible impact on conventional diffraction limited

imaging, they can become significant in SMLM where nanometric resolution are targeted.

The answer provided by the authors is very general and does not address my question regarding the sample. To reiterate: Is there z-drift induced by evaporation of the immersion water? And/or is there sample drift due to contraction/expansion of the hydrogel, or something else? I understand that the microscope setup will be subjected to some (small) drift due to the lack of temperature stabilization and vibration isolation in the room.

(Remarks on code availability)

REVIEWER COMMENTS

Reviewer #1

The authors describe soSMART, an approach where they add embedded fiducial beads at various heights to their previously published soSPIM approach (Galland et al. 3D high- and super-resolution imaging using single-objective SPIM. *Nat Methods* 12, 641–644 (2015)) to enable real-time drift correction, adaptive optics for correction of depth-induced aberrations, and simplified stitching of multiple image slices. The acquisition pipeline is then automated for a simplified workflow.

Their approach is useful to the field and should in principle improve long-term imaging by the active drift stabilization, the achievable precision and resolution through the adaptive optics, and the ease and time required for slice stitching. Their approaches are mostly well-described and most claims are supported by their data. However, the authors should address the following points to clarify the performance of their system, compare their approach to previously published approaches, and enable reproducibility.

1. The authors should include comparisons to previous approaches for live drift correction used for single-molecule imaging at depth. Some of these are shown in SI Fig. 1, but the pros and cons of these approaches are not clearly outlined. For example, fiducial-free drift correction has been demonstrated for 3D SMLM (Wester et al. Robust, fiducial-free drift correction for super-resolution imaging. *Sci Rep* 11, 23672 (2021)). Also, separate detection of non-fluorescent beads can be used for 3D drift correction over an extended depth range (Coelho et al. Ultraprecise single-molecule localization microscopy enables in situ distance measurements in intact cells. *Sci. Adv.* 6, (2020)). This approach was recently demonstrated to work well in a similar light sheet work, while providing better lateral and axial precision (Saliba et al., Whole-cell multi-target single-molecule super-resolution imaging in 3D with microfluidics and a single-objective tilted light sheet, *Nat. Commun.*, 15 1-17 (2024)). This work also used a single-objective light sheet, active drift stabilization, and DECODE for tenfold improvement in acquisition speed for 3D whole-cell DNA-PAINT imaging, so the authors should compare their design to this work and add it to SI Fig. 1.

We thank the reviewer for highlighting these references, which address the challenging task of single molecule imaging in depth at the entire cell scale. Our primary intention with **SI Fig. 1** and the related discussion was to provide a general overview of the different methods proposed in the literature for in-depth SMLM over axially extended volumes, and highlight the main challenges to address. We did not initially intend to provide an exhaustive comparison of the pro and cons of each method, but following your suggestion, we have now incorporated it in the revised version of the manuscript, as it can be useful for the readers.

We would like to note that the work by Saliba et al. was not yet published at the time of our manuscript submission. We have now included it in the revised version of the manuscript and discussed its specific contributions in relation to the objectives of our study.

2. In terms of both live drift correction and slice stitching, the pros and cons of long axial range point spread functions should be discussed. These can be used to cover the whole axial range of cells, as shown in e.g. (Nehme et al. DeepSTORM3D: dense 3D localization microscopy and PSF design by deep learning. *Nat Methods* 17, 734–740 (2020)) and (Nakatani et al., Long-axial-range double-helix point spread functions for 3D volumetric super-resolution imaging, *J. Phys. Chem. B*, 128 (2024)).

We thank the reviewer for this comment. We agree that long axial range PSFs, such as DH-PSF used in Nakatani et al. (also published after our submission) or the tetrapod-PSF employed in Nehme et al., allow imaging volumes up to 4 to 5 μm in height without the need for sequential plane-by-plane acquisition. However, these approaches do not eliminate the need for multiple overlapping acquisitions when imaging larger volumes (10 to 15 μm in height), which is the primary goal of our work.

However, PSF-shaping approaches are highly sensitive to optical aberrations. While aberrations at the proximity of the coverslips (as in Nehme et al., Nakatani et al. and Saliba et al.) are limited, imaging deeper, at several tens of microns above the coverslips, presents another challenge which we addressed using an Adaptive Optics correction strategy. Moreover, imaging at such depths precluded the possibility of using fiducials adsorbed at the coverslips, even with 10 μm long-axial range PSFs.

Another important limitation of long axial range PSFs is their much bigger sizes, which inherently increase the single-emitter density. This makes the use of advanced deep learning-based localization algorithms even more essential, and still imposes constraints, which is not yet turn-key.

We have included a detailed discussion of these points in the revised version of the manuscript.

3. Regarding AO, the authors should discuss their work in context with previous similar work using AO together with a single-objective light sheet (Hung et al., Adaptive optics in single objective inclined light sheet microscopy enables three-dimensional localization microscopy in adult *Drosophila* brains, *Front. Neurosci.*, 16 (2022)). Here the authors should discuss why their approach is better than the in situ calibration of PSFs from the single-molecule data itself used by Hung et al: INSPR (Xu, F. et al. Three-dimensional nanoscopy of whole cells and tissues with in

situ point spread function retrieval. *Nat. Methods* 17, 531–540 (2020)), as well as other approaches such as DL-AO (Zhang, P., Ma, D., Cheng, X. et al. Deep learning-driven adaptive optics for single-molecule localization microscopy. *Nat Methods* 20, 1748–1758 (2023)), VISPR (Yang et al., "Accurate 3D single-molecule localization via vectorial in situ point spread function retrieval and aberration assessment," *Photon. Res.* 12, 2447-2461 (2024)), etc.

We thank the reviewer for raising this point. One key advantage of our approach is the direct measurement of the experimental PSF, before and after AO correction, thanks to the fiducials embedded in the SMART devices.

PSF retrieval methods, such as the in situ PSF calibration from single-molecule data used by Hung et al. (INSPIRE) or other approaches like DL-AO (Zhang et al., 2023) and VISPR (Yang et al., 2024), are indeed valuable solutions, but they are still more complex to implement correctly. Additionally, they are sensitive to the single molecule density and signal to noise ratio, which can limit their reliability in certain experimental conditions. In contrast, fiducial markers are ideal for direct experimental PSF measurement as they are isolated, photostable and bright.

We have expanded our discussion on these aspects in the revised manuscript, highlighting both the advantages of our approach and the potential for future work that could require the use of PSF retrieval techniques.

It would also be useful if the authors could demonstrate and quantify that using nanodiamonds outside the well for AO successfully corrects the PSFs of the single-molecule data in the sample, where there will be additional sample-induced aberrations present. It would also be useful for the authors to discuss effects of field-dependent aberrations, especially in the context of imaging large fields of view as shown in Fig. 3B.

Indeed, our method primarily corrects for the microscope- and depth-induced aberrations, and does not account for sample-induced aberrations. However, we believe that these sample-induced aberrations can be considered negligible in the case of single cell or small cells aggregate. This is further supported by the fact that none of the SMLM approaches for imaging entire cells account for sample-induced optical aberrations, as summarized in **SI Fig. 1**. In our case, higher depth imaging made it important to correct for system- and depth- induced aberrations. We have clarified this point in the revised manuscript and discussed the future requirements for imaging more complex samples that will induce additional aberrations.

Regarding the validity of the optical aberration correction using nanodiamonds outside of the well, we were unable to perform a direct comparison between PSFs outside and inside the well due to a lack of direct access to PSFs within the sample volume. Nonetheless, previous studies (Han et al., *A polymer gel index-matched to water enables diverse applications in fluorescence microscopy*, *Lab On a Chip* 21(8), 1549-1562 (2021); and Ravasio et al., *High-resolution imaging of cellular processes across textured surfaces using an indexed-matched elastomer*, *Acta Biomaterialia* 14, 53-60 (2015)) have shown that the water index-matched polymer used introduces negligible to no additional aberrations, apart from depth-dependent aberrations, even through layers as thick as 150 μm . These results, which we now cite in the revised manuscript, support the validity of our approach to correct aberrations using nanodiamonds located close to the well, at the imaging depth where depth-dependent aberrations are dominant.

Finally, we acknowledge that field dependent aberrations may become more significant when imaging large fields of view and complex samples. Following your suggestion, we have discussed this point in the revised version of the manuscript, specifically in the context of the results shown in **Fig 3B** as the field of view in our other acquisitions were small enough in regards to field dependent aberrations.

4. The authors should discuss the relatively large localization uncertainty of their nanodiamonds used for live drift correction (29.6 nm and 14.6 nm axially and laterally, respectively), both in terms of why the uncertainty is this large compared to the typical localization precision of conventional fluorescent beads, and any measures you need to take, e.g. averaging etc, to not scramble/convolve your single-molecule data with this uncertainty in fiducial localization. The authors do briefly describe in the Methods that temporal averaging can be selected to improve the precision, but it would be useful to discuss the effect and need for this more in the main text.

We thank the reviewer for pointing out this issue. It simply comes from a mistake from our side between pointing accuracy and resolution, which has now been corrected in the revised manuscript.

We also better detailed in the main text the primary goal of our active drift correction system, which is to maintain the alignment of the excitation and detection planes throughout the acquisition process. This does not require nanometric correction precision, as an additional off-line drift compensation is performed to ensure optimal correction for the entire volume reconstruction.

Even if not required, we agree with the reviewer that temporal averaging can improve the accuracy of drift compensation during the acquisition. We have made this point clearer in the revised manuscript.

5. In Fig. 3B, the authors demonstrate imaging of 3D cell cultures. However, for these measurements, the new SMART devices and their beneficial aspects (embedded fiducial beads for real-time drift correction, AO, and simplified slice stitching) were not used, so these data seem disconnected from the main story of the manuscript. Also, the authors do not demonstrate how well this imaging performs compared to the data shown in Fig. 2 in terms of precision and FRC resolution.

We acknowledge that the data presented in **Fig. 3B** do not utilize the soSMART devices. We mostly aimed with this acquisition to demonstrate the versatility of the soSPIM imaging architecture and its potential to image several

tens of microns within thicker biological samples such as spheroid and organoid. To our knowledge, this has not yet been demonstrated with other light-sheet based SMLM systems. These results also serve to open the discussion on new applications and needs for alternative methods for drift compensation and AO correction, in absence of fiducials and within samples inducing aberrations. Given this, and in light of Reviewer 3's interest in these applications, we propose to retain these results in the manuscript.

We indeed didn't include a direct measure of the imaging performance for this sample. This is now addressed in the revised manuscript using FRC resolution.

The data presented in Fig. 3A is simply supporting previous studies showing about a tenfold speed improvement when using DECODE (and other deep-learning-based approaches) compared to fitting-based analyses, see e.g. (Speiser et al. Deep learning enables fast and dense single-molecule localization with high accuracy. *Nat Methods* 18, 1082–1090 (2021)), (Saliba et al., Whole-cell multi-target single-molecule super-resolution imaging in 3D with microfluidics and a single-objective tilted light sheet, *Nat. Commun.*, 15 1-17 (2024)), and (Nakatani et al., Long-axial-range double-helix point spread functions for 3D volumetric super-resolution imaging, *J. Phys. Chem. B*, 128 (2024)).

The reviewer is correct. The data presented in **Fig. 3A** aims to demonstrate that deep learning-based high density localization approaches, such as DECODE, can address a key challenge in volumetric SMLM. We implemented DECODE in direct collaboration with J. Ries, who codeveloped the method.

This is indeed similar to the works by Saliba et al. and Nakatani et al., which were not yet published when we submitted our manuscript to Nature Communications. The revised manuscript now cites these works, reinforcing the argument concerning the need of such computational approaches in the future for extended volume 3D SMLM in routine.

6. On line 86, the authors state that the performances of 3D cross-correlation approaches have not yet been demonstrated. The authors may want to rephrase his statement, as 3D cross-correlation has been used e.g. for slice stitching (Saliba et al., Whole-cell multi-target single-molecule super-resolution imaging in 3D with microfluidics and a single-objective tilted light sheet, *Nat. Commun.*, 15 1-17 (2024)).

We agree that the work of Saliba et al., published after the submission of our manuscript, demonstrates the use of 3D cross-correlation for slice stitching, provided there is sufficient overlap between the sequentially acquired planes. However, in our case, using shorter axial-range astigmatism PSFs would have required acquiring a larger number of slices, which is why 3D cross-correlation was not an optimal solution for our setup. We have rephrased this statement and discussed its implications in the revised manuscript.

7. On line 111-112: The authors state that they achieve resolution of 7 nm laterally and 40 nm axially. Please clarify if this is actually resolution (as from FRC calculations) or localization precision. None of their shown FRC curves show a 7 nm resolution. Please also clarify on line 241.

The values refer to the resolutions measured from the localization precision of embedded nanodiamond into the soSMART device, using $r \approx 2.35 \times \sigma$, where σ is the nanodiamonds localization precision. We have clarified this in the revised manuscript and in the Methods.

8. On line 137 and throughout: please match the terminology between the main text and the figures. On line 137, the authors mention third and fifth order spherical aberrations, whereas in e.g. SI Fig. 2 the authors refer to primary and secondary spherical aberrations.

We thank the reviewer for pointing out this discrepancy. We have corrected the terminology in the revised manuscript. We now refer to "third and fifth order spherical aberrations" in the main text (line 137) and **SI Fig. 2**.

9. Throughout the manuscript, please make sure to describe what errors indicate, e.g. standard deviation or SEM, and the number of measurements (n). See e.g. lines 175, 179, and 604.

We have clarified these values in the revised manuscript, and indicated the number of measurements (n) when applicable.

10. On lines 202, 228, 241, and 273: The authors provide a single resolution estimation (31 nm, 55 nm, 78 nm, and 42 nm, respectively). Please state if this is estimated in full 3D (i.e. FSC), or which plane this refers to (e.g. xy). Importantly, the authors should provide the FRC resolution values both laterally (xy) and axially (xz and/or yz) for all data throughout the manuscript, including the data shown in Fig. 3B, to demonstrate the achievable performance of their system for various samples.

On line 202, the FRC value corresponds to the lateral FRC (xy) computed from the sum projection of all localizations acquired on a single image plane. For all the other FRC measurements (summarized in **SI Fig. 8**), the values correspond to the mean and s.e.m values of lateral FRC values computed for each plane of a reconstructed volume, every 200 nm in the z-axis, as detailed in the Methods section. This has been clarified in the revised manuscript.

However, we were unable to compute the axial FRC resolution for our various acquisitions, as there are not freely available solutions. The axial FRC value presented in the work of Saliba et al. were computed using the commercial SRX Vutura software.

We would also like to emphasize that while FRC computation provides a useful estimate of image resolution, it is inherently limited in its ability to detect reconstruction artifacts, as long as these artifacts are reproducible. Indeed, we collected new dataset to demonstrate the benefits of aberration correction in our system for imaging cells located tens of microns above the coverslips (**SI Fig 6**). We observed that depth-dependent aberrations significantly degrade PSF quality, leading to reduced localization accuracy and the emergence of reconstruction artifacts. In particular, we noted the presence of echoes, characteristics of spherical aberrations when using 2D Gaussian fitting-based localization methods. Despite these visible artifacts in panel A and B (without aberrations corrections), the FRC yielded similar resolution estimate for the three conditions (without and with aberrations corrections): $88 \pm 9 \text{ nm}$, $103 \pm 19 \text{ nm}$ and $103 \pm 19 \text{ nm}$ for panels A, B and C respectively (*mean \pm s.e.m., n = 5*). These results highlight the limitations of FRC in fully capturing image fidelity in the presence of optical aberrations.

11. In the Results section, the authors mention that the data is filtered in post-processing. Please report all filters used.

The details of the filters used for the reconstruction process were already provided in the “Whole cell reconstruction” section of the Methods. We applied the same filters to all acquisitions. We clarified this in the revised manuscript.

12. For the lamin B1 reconstruction in Fig. 1, the authors report a very different thickness (79 nm vs 182 nm) in the lateral and axial direction. Is the lamina expected to vary that much in thickness at various parts of the cell, or is this a result of the different resolution achieved laterally and axially, convolving the measured thickness? The authors should discuss this.

The reviewer is right. The difference in thickness (79 nm vs. 182 nm) in the lateral and axial directions is indeed the result of the different resolutions radially and axially. We have clarified this point in the revised version of the manuscript.

13. On line 203, the authors likely refer to Fig. 1E rather than Fig. 1D.

We thank the reviewer for pointing out this mistake. It has been corrected in the revised version of the manuscript.

14. For the suspended Jurkat cell imaging in the Results section (starting at line 243), please indicate at what height or height range above the coverslip these cells were imaged at.

In general, all the cells in the manuscript (COS7 and Jurkat cells) were located between 10 and 35 μm above the coverslips. We have now clearly stated this information in the revised manuscript and in the Figures.

15. On lines 260-261, the authors provide three lateral and three axial cluster sizes, but only specify two targets (PD-1 and CD3). Please clarify.

We provided three lateral and three axial cluster sizes from three independent acquisitions (soSMART devices, cell seeding and acquisition), with one measurement for PD-1 imaging and two measurements for CD3 imaging. We have clarified this in the revised manuscript.

16. Lines 283-284: When discussing faster alternative docking and imager DNA sequences, the authors should also include the work by (Chung et al. Fluorogenic DNA-PAINT for faster, low-background super-resolution imaging. Nat Methods 19, 554–559 (2022)).

We thank the reviewer for highlighting another work for fast DNA-PAINT imaging. We have added this reference in the discussion.

17. Please provide the used values or ranges for the DECODE training parameters (lines 764-765).

We have added all the training parameters for DECODE in the Methods section.

18. It would be useful to add a discussion about live-cell compatibility of their system for readers interested in using this approach for live-cell studies in the future.

We thank the reviewer for this suggestion. We now have added a discussion on the live-cell compatibility of our system, as already demonstrated in (Galland et al., “soSPIM: single-objective Selective Plane Illumination Microscopy for 3D high and Super-resolution imaging of biological structures.” Nature Methods, 2015, 12(7), pp 641-44) and in (Aoun et al., “Amoeboid swimming is propelled by molecular paddling in lymphocytes”, Biophysical Journal, 2020, 119, 1157-1177).

19. Please add a schematic of the optical setup as a new SI Figure. The authors mention that the soSPIM beam steering unit was derived from their Galland et al. paper, but there are notable changes, such as the optics used to conjugate the objective back focal plane onto the deformable mirror for AO.

Thanks for the suggestions. It is added in the SI figures of the revised version of the manuscript (**SI Fig. 13**).

20. Also please add a schematic of the full fabrication pipeline. This is a key component of the paper, and there are differences between this fabrication pipeline and that which was published in their Galland et al. paper.

Thanks for the suggestions. It is added in the SI figures of the revised version of the manuscript (**SI Fig. 14**).

21. Please clarify the need for using nanodiamonds compared to regular fluorescent beads in the main text. The authors briefly touch upon this in the Methods, but this is an important consideration to discuss.

We prefer nanodiamonds over fluorescent beads due to their superior photostability. The long acquisition times required for SMLM and the UV illumination involved in the SMART devices fabrication steps make photostability of the fiducials essential. They are also dimmer than fluorescent bead, such as tetraspeck, which often saturate in SMLM illumination conditions, preventing their use for registration. This has been clarified both in the main text and in the Methods section of the revised manuscript.

22. The authors mention that COS7 cells were fixed 60 minutes after seeding. Please include a motivation for this relatively short time frame, as the cells have limited time to attach during this time. This is particularly important to mention if the reason is that the chambers are not compatible for long term live-cell work.

We thank the reviewer for raising this point. The relatively short time frame before fixation was chosen to image suspended cells displaying wrinkled shaped-nuclei, as they display more features and complexity to images compared to smooth nuclei. However, cells can be easily cultured for several days within the soSMART devices before imaging (See Fig. 2 in Reviewer #2 response and new SI Fig. 10 where cells were cultured for 48h prior fixation and labelling). We clarified this point in the revised manuscript and have added a discussion of the live compatibility of our method.

23. In the methods, the authors mention that the index matched polymer is illuminated with UV for two hours to bleach the polymer auto-fluorescence. Please include the intensity of the light used.

It has been included in the revised manuscript in the Methods section.

24. Please include the camera settings used for all measurements.

They have been included in the revised manuscript.

25. Please include the rendering settings used for all reconstructions.

They have been included in the revised manuscript in the Methods section.

26. Please include the computer and GPU specifications used for the DECODE training and analysis for reproducibility.

They have been included in the revised manuscript.

27. Please include the sequences and/or the order number and company for all used DNA PAINT kit.

They have been included in the revised manuscript.

28. For all main and SI figures with 3D reconstructions, it would be useful if the authors specify if the number on the colorbars indicate the full z range, or they can add z-ticks to more clearly show the z values. In Fig. 2, please indicate the imaged z-range for each panel. SI Fig. 10 lacks the z indication.

We thank the reviewer for this suggestion. We have indicated the depth at which each cell has been acquired (between 10 to 35 μm from the coverslip) in the "Acquisition and reconstruction parameters" table and in the main and supplementary figures.

29. SI Fig. 1: Please increase the size and font size to make it more legible.

We made efforts to ensure that all figures, and especially SI Fig. 1, are more legible.

30. SI Fig. 2: Please state what the error bars indicate. Also, on line 37, please describe what the error indicates ($n = 84 \pm 19$).

It has been clarified in the revised version.

31. SI Fig. 3: Please add zoom in on a small part of the y-axis to visualize drift and jumping when correcting. Please quantify the stability for the two smart methods. From here, it's difficult to say if XY + PSF or XYZ is the best, and how they differ from just PSF. Also please include if averaging was used for correction or if frame-by-frame correction was used, for each data set shown.

The PFS (perfect focus system) of the microscope has a similar resolution (slightly worse) for axial drift correction compared to our fiducial-based method. However, it has a limited axial working distance ($\approx 40 \mu\text{m}$), and sometime interferes with the soSMART mirrors, which affects its locking capability. Moreover, it can only correct for axial drift (z-axis), while in the soSMART configuration there is also a need to correct for lateral drifts (x-, y-axis) to keep the light sheet overlaid with the focal plane. This is clarified this in the revised version of the manuscript.

Regarding the drift and jumping, we performed frame-by-frame real-time drift correction, which ensures a smooth correction process. This is clarified for each dataset shown in the SI fig. legend and the Methods section.

32. SI Fig. 4: Please show also with adjusted contrast to show the cells, which cannot be seen in the current version.

Thanks for the suggestion. We adjusted the gamma and the contrast to better visualize the cells inside the wells in SI Fig. 4.

The bead densities noted here are different from the density noted in results (4.5 ± 3.1) $\times 10^{-3}$. Please clarify or explain the range.

The bead density mentioned in the Results (line 190) section of the manuscript is the $mean \pm s. e. m.$ over 4 different acquisitions made on 4 different soSMART devices as stated in the methods section (line 614). The density in **SI Fig. 4** corresponds to the density of the soSMART device used for these two acquisitions. This difference is due to the pipetting variability and high viscosity of the bio-compatible polymer.

33. SI Fig. 8: Please add this data for all image acquisitions presented, and show the FRC calculations both laterally and axially.

We thank the reviewer for the suggestion. We added the nanodiamonds localization precision and lateral FRC computations for all the data presented in the work. However, as mentioned earlier, we were unable to compute the axial FRC resolution, as no freely available solution exists, and the computation in Saliba et al. was done using commercial software (SRX Vutura), which we do not have access to.

34. Code availability: please make the SMARtrack and soSPIM plugin codes and the custom ImageJ macro used to reduce grid artefacts from DECODE-analyzed data available on e.g. GitHub.

The SMARtrack plugin, home-made code and installers will be made available on GitHub repository.

35. Data availability: please make the source data presented in this manuscript available through e.g. Zenodo.

The source data will also be made available through Zenodo.

Reviewer #2

Cabillic et al. report a new 3D SMLM technique that allows high SNR, volumetric super-resolution imaging of cells and organoids with extremely high stability. For that, the authors extended their previous approach—single objective selective plane illumination microscopy (soSPIM)—where microchambers with reflective surface provides thin illumination. Unlike soSPIM, a new approach introduced fluorescent nanodiamonds around the chamber which enables depth-dependent aberration correction and precise 3D drift correction. These issues are quite challenging to be solved but the authors demonstrated its feasibility with rigorous and quantitative analysis. The reviewer could clearly see the potential of this approach in the bioimaging community. However, a few major concerns were found, which needs to be clearly addressed before publication.

1. The first concern starts from laminB1 structure of COS-7 Cells in Fig. 1E and Fig. 2B. For healthy condition, laminB1 should show oval, wrinkle-free shape but all the images display nucleus wrinkling, depicting abnormal phenotype. This is likely due to the fact that COS-7 cells were not able to properly adhere to the surface and/or to communicate with other cells. Considering the physical dimension of the microfabricated device, it is unlikely that the chamber accommodates multiple COS-7 cells. In contrast, for Jurkat T-cells, the distribution of membrane proteins looks normal because they are floating cells. It means that soSMART would not be suitable for adherent cells.

We thank the Review for his positive and constructive feedbacks. This is correct that under healthy conditions, the laminB1 structure should appear oval and wrinkle-free. However, we intentionally imaged COS-7 cells with wrinkled and invaginated nuclei to better illustrate the resolution capacity of our super-resolution imaging method. For this aim, cells were fixed shortly after seeding, before fully adhering to the well walls. We acknowledge that this condition is not physiologically relevant, and we have clarified this in the revised version of the manuscript.

To demonstrate that cells can indeed adhere to the wells, we acquired additional super-resolution data from COS7 cells cultured for 48 hours in the SMART devices prior fixation, labelling and imaging (see below). These cells exhibited the expected oval, wrinkled-free shape of the laminB1 structures of adherent cells, which are spread onto the walls of our wells. We have also included this figure in the revised manuscript (**SI Fig. 10**) to better illustrate the cell culture and live imaging compatibility of the SMART devices.

Figure 1: Volumetric 3D-SMLM of COS7 cells cultured for 48h in a soSMART device (SI Fig. 10). (A) 3D soSMART acquisition of the Lamin B1 nuclear envelop of two COS-7 cells in a well of a SMART device. Left: 1.65 μm thick reconstructions at different z positions. Right: 10 μm thick 3D reconstruction of the entire volume. Color codes with the z-distance to the coverslip. **(B)** Schematic representation (top: side view; bottom: top view) of the position of the COS-7 cells adhering to the walls of a SMART device. **(C)** Spatial resolution estimation for this acquisition (mean±s.e.m.) of the FRC spatial resolution computed on each reconstructed plane of the entire volume (n=50).

Regarding the live cell capability, our devices have already been shown to be compatible with live-cell imaging in (Galland et al., “soSPIM: single-objective Selective Plane Illumination Microscopy for 3D high and Super-resolution imaging of biological structures.” Nature Methods, 2015, 12(7), pp 641-44), (Singh et al., “3D Protein Dynamics in the Cell Nucleus”, Biophysical Journal, 2017, 112, pp 133-42) and (Aoun et al., “Amoeboid swimming is propelled by molecular paddling in lymphocytes”, Biophysical Journal, 2020, 119, 1157-1177). The results from these studies, summarized in the figure 2 below, demonstrate that our devices are suitable for cell culturing and live-cell imaging. These studies used wells made of the same biocompatible polymer, without the embedded inert nanodiamonds present in our current device.

We included a discussion about the live compatibility in the revised manuscript.

[editorial note: figures redacted]

Figure 2: soSPIM live cell compatibility. (I) *SI Fig. 6* of the paper Galland et al, 2015, illustrating the possibility to culture and image adherent cells within the soSPIM devices. (II) *Fig. 5* of the paper Aoun et al, 2020, illustrating the capability of the soSPIM technique to image light-sensitive cells cultured in the soSPIM devices. (III) *Fig. 4* of the paper Singh et al, 2017, illustrating the possibility to record the dynamics of nuclear protein through imFCS within soSPIM device using the soSPIM imaging capacity.

Regarding the dimensions of the wells hosting the cells in the microfabricated devices, we demonstrated in (Galland et al., “soSPIM: single-objective Selective Plane Illumination Microscopy for 3D high and Super-resolution imaging of biological structures.” *Nature Methods*, 2015, 12(7), pp 641-44), that wells can be fabricated with various dimensions: $22 \times 22 \mu\text{m}^2$, $24 \times 24 \mu\text{m}^2$, $40 \times 40 \mu\text{m}^2$ and $60 \times 300 \mu\text{m}^2$, all with a height of $40 \mu\text{m}$. These designs were made to accommodate isolated single cells, cells doublets up to cell aggregates. In the present study, we used $40 \times 40 \times 40 \mu\text{m}^3$ wells, which can easily host multiple COS 7 or Jurkat cells. We have added a description of these design options in the revised manuscript to better highlight the versatility of our technique.

2. A follow-up question is that it was hard to figure out how suspended COS-7 cells were immobilized on the chamber. Is their basal surface partially attached to the surface? Brightfield images would be helpful to confirm that.

Regarding the immobilization of suspended COS-7 cells, we chose cells that were fixed before they fully adhere. These cells displayed protrusions extending toward the edge of the wells, which prevented them from moving. Allowing them more time to adhere, the cells would have spread onto the nearest surface within the well, as shown in **Fig. 1** and **Fig. 2I**. We did not use any specific coatings (e.g., fibronectin, laminin, ...) or non-adhesive coating (e.g., Pluronic, Lipidure) in our study to functionalize or passivate the wells. While we acknowledge that brightfield images would be helpful to confirm this immobilization, we did not collect such image for the different acquisition shown in this study.

3. The next concern is the layout of the manuscript. It has two separate stories—imaging cells by soSMART and imaging organoids by JeWell + soSPIM. I fully understand why the authors wanted to include the latter but it's technically different. It didn't exploit fiducial nanodiamonds which is the core of the manuscript, I believe.

We thank the reviewer for pointing this out and agree that the data presented in **Fig. 3B** do not utilize the SMART devices. However, our intention was to showcase the versatility of the soSPIM imaging architecture and its ability to image several tens of microns within thicker and more challenging biological samples, such as spheroids and organoids. To the best of our knowledge, no other light-sheet based SMLM system has demonstrated this capability. Including this data also expands the discussion to new applications and highlights the need for alternative methods for drift correction and AO correction in absence of fiduciary markers. Moreover, given Reviewer 3's interest in

these broader applications, we propose retaining this figure as it opens a whole new field of applications, as well as ongoing and future developments.

There are also minor comments below:

1. Light sheet thickness and length need to be reported

We have added a supplementary figure describing the light-sheet thickness and length in the Methods section of the revised manuscript.

2. Are there any shadow effects on the excitation beam due to the embedded nanodiamonds?

Nanodiamonds, with a diameter of 70 nm, are too small to induce noticeable shadowing effects on the excitation beam. Furthermore, the digital scanning approach used to create the light-sheet mitigates potential shadowing that could arise.

3. It is crucial to show 3D DNA-PAINT imaging without and with AO correction. Supplementary Figure 2E is not sufficient to see the effect.

We thank the reviewer for this suggestion. We initially did not collect 3D DNA-PAINT datasets without AO correction, as we aimed to show optimal SMLM reconstructions using an appropriate Gaussian fitting model. We have now collected new datasets to demonstrate the benefit of AO correction for 3D single molecule localization in depth in a new **SI Fig. 06**. This figure clearly shows that depth-dependent aberrations significantly degrade PSF quality, leading to reduced reconstruction accuracy and the appearance of reconstruction artifacts, regardless of whether the astigmatism was calibrated at the coverslips or at the cells depth. Correcting these depth-dependent aberrations with our DM effectively restored PSF quality, resulting in accurate 3D reconstruction (**Supp. Fig 6 C**). This supplementary figure has been included in the revised version of the manuscript with a discussion in the main text.

4. Several typos/errors were found for example:

Line 90: ref30 is incorrect.

Line 225: 400 um  400 nm

Line 443: 700 nm  700 mm

Supplementary ref1 needs to be switched with ref2.

We thank the reviewer for pointing these errors out. They have been corrected in the revised version of the manuscript.

Reviewer #3

This manuscript presents an improved single molecule localization microscopy for deeper biological process investigation. The authors used microfabricated devices for light-sheet microscopy, adaptive optics for aberration correction, and real-time feedback for drift correction. The approach was applied to assess 3D nanoscale protein organization in (fixed) cells. Overall, the manuscript is well written and seems scientifically sound.

Comments:

Figs. 1-3: Regarding the color coding of the position along the z-axis: To my understanding the value in micrometers printed on the color bars indicate the ranges from the blue to the red color. I think it would help the reader to instead overlay the color bars with axes with ticks indicating a sequence of corresponding z positions, e.g., 1 um, 2 um, 3 um, etc.

We thank the reviewer for the positive and constructive feedbacks. Regarding the z-color coding, we have clarified all the reconstructions in the revised manuscript by overlaying the color bars with the corresponding z-ranges, making it easier to read the acquired volumes and imaging distances.

Drift. The authors went to great lengths to correct for drift during the acquisition (which is a good thing). Yet, to me it is unclear what the (main) sources of drift are. Is it mechanical drift of the microscope? Z-drift induced by evaporation of the immersion water? Or sample drift due to contraction/expansion of the hydrogel, or something else? It would be good to understand and discuss the underlying causes. For example, if the evaporation of immersion water over several hours of imaging is a problem, some immersion oils with water-like refractive index could be used.

Drift is a common issue in microscopy. It is particularly visible and negative in Single Molecule Localization based Super-Resolution Microscopy, due to the long acquisition times (ranging from minutes to hours) and the aim for nanometric resolutions. Main sources of drift in microscopy are likely due to thermal fluctuations causing the contraction and expansion of components such as the microscope or stage, as well as potential vibrations from the

microscope and camera. While these factors usually have a minimal visible impact on conventional diffraction-limited imaging, they can become significant in SMLM where nanometric resolution are targeted.

Fig. 3B iii. This figure panels shows a typical challenge with light sheet microscopy: The cells on the right side, which are further from the incident light sheet, are poorly resolved because the light sheet is distorted and scattered passing through the cells on the left (close to the incident light sheet). Would it make sense to use self-reconstructing Bessel or other (non-Gaussian) beam shapes? Or dual illumination from both sides? I understands this complicates the experimental setup, but I think these challenges should be at least mentioned and discussed.

Thank you for this insightful comment. Indeed, this is a common issue in single-side illumination light-sheet microscopy with Gaussian beam, which can be addressed by using either non-diffracting beam (e.g., Bessel, Airy, ...) or Multiview approaches.

Non-diffracting beam generally require a particularly high NA to produce a thin and long light-sheet. This would limit the axial scanning range in the soSPIM geometry, due to the larger beam diameter that would need to be reflected onto the 45° mirror.

The truncated pyramidal-shaped well used in Fig. 3B features four 45° mirror surrounding the sample. These four mirrors provide access to the other side of the sample without passing through it, but imaging would still need to be performed sequentially. We consider this multi-view capability to be beyond the scope of the current manuscript, but we have discussed it in the revised manuscript.

Fig. 3B ii&iii. Somehow the inset of ii) magnified in iii) looks different. Maybe it is because of the color code rescaling to a different range? It would be good to show the data in the same way for consistency.

We thank the reviewer for pointing this out. Indeed, the inset in panel ii and iii of Fig. 3B do not display the same axial range. We have clarified this by systematically specifying the absolute z-range of the different volumes, as indicated by the corresponding color bar in all figures when relevant. This ensures consistency in the presentation of the data.

Live cell usability. If I understand correctly, all data shown were acquired in fixed cells. It would be fantastic if the 3D-SMLM approach shown in this manuscript could be applied to live cell samples as well. The long acquisition times are of course challenging and the required temperature control to typically 37C will accelerate any drift. It would be good if the authors could discuss if and how this method could be applied to live samples.

We thank the reviewer for the thoughtful comment. We invite the reviewer to refer to our response to Reviewer #2 regarding the live-cell compatibility of our method. We have addressed this topic in more detail in the revised manuscript, where we discuss the potential of our method for 3D-SMLM on live sample.

Figure readability. Some of the fonts are way too small, for example the scale bar labels in Fig 3. While a PDF copy allows to zoom in, the authors should ensure to maintain readability when printed out on A4 paper.

We thank the reviewer for this comment. We made efforts to ensure that all figures are legible when printed on A4 paper.

REVIEWER COMMENTS

Reviewer #1

Overall, the authors have made substantial revisions that have further strengthened the manuscript. They have satisfactorily addressed all but one of my previous comments.

7. The authors still refer to the localization precision of the nanodiamond data as “resolution”, and report the resulting localization precision as resolution in the same sentences as when discussing the FRC resolution of the single-molecule data (see e.g. lines 125-126, 270, 284, 301-303, and 327). This is very likely to confuse the readers. Particularly, on line 125-126, the useful metric to report is the achievable FRC resolution from the single-molecule data rather than the achievable localization precision of nanodiamonds, and this should be changed. The authors more clearly state this distinction in their SI Fig. 8, where they state “Beads localization precision” and “FRC per plane”.

We thank the reviewer for pointing out this confusion. Indeed, the values extracted from the localization of fiduciary markers were computed as the FWHM of the localizations distribution for each marker, which is commonly interpreted as an estimate of the system’s minimum achievable resolution. We have corrected this terminology in the revised version of the manuscript, and now refer only to resolution and not pointing accuracy. Additionally, SI FIG. 8 has been updated to present these measurements as “beads FWHM measurements”, in line with the reviewer’s recommendation and consistency with the figure’s legend.

However, we would like to retain both the FRC and the FWHM derived from fiduciary markers’ localization when applicable, as we believe they provide complementary information about the overall resolution and quality of the SMLM data. Indeed, FRC is a valuable metric that combines localization precision, density, and structural information, but it does not account for potential localization artifacts (see previous response 10 to reviewer #1 and SI Fig. 06. Conversely, FWHM of single-point emitter localization provides a useful raw estimate of the resolution at which individual molecules are detected.

I also strongly encourage the authors to make the differences between the data acquisition approaches in Fig. 3B vs the rest of the manuscript very clear (pointed out by both reviewer 1 (comment #5) and reviewer 2).

We clarified this important difference by removing all mentions of the soSMART method or SMART devices from the corresponding section. We also discussed how drift and aberration correction could be overcome in absence of fiducials. To make the difference clearer, we also split the original Fig. 3 (DL-based acceleration and 3D culture imaging) in two distinct figures. The optimization of the soSMART acquisition speed using a deep-learning based localization approach is now presented in the revised Fig. 3, while the demonstration of the soSPIM imaging capabilities in complex samples is presented separately in the new Fig. 4.

Reviewer #2 (Remarks to the Author):

The authors have clearly addressed all the concerns that I had. I don't have any further comments.

Reviewer #3 (Remarks to the Author):

In their revised manuscript, the authors have reasonably addressed my questions except one:

Question: The authors went to great lengths to correct for drift during the acquisition (which is a good thing). Yet, to me it is unclear what the (main) sources of drift are. Is it mechanical drift of the microscope? Z-drift induced by evaporation of the immersion water? Or sample drift due to contraction/expansion of the hydrogel, or something else? It would be good to understand and discuss the underlying causes. For example, if the evaporation of immersion water over several hours of imaging is a problem, some immersion oils with water-like refractive index could be used.

Answer in the rebuttal: Drift is a common issue in microscopy. It is particularly visible and negative in Single Molecule Localization based Super-Resolution Microscopy, due to the long acquisition times (ranging from minutes to hours) and the aim for nanometric resolutions. Main sources of drift in microscopy are likely due to thermal

fluctuations causing the contraction and expansion of components such as the microscope or stage, as well as potential vibrations from the microscope and camera. While these factors usually have a minimal visible impact on conventional diffraction limited imaging, they can become significant in SMLM where nanometric resolution are targeted.

The answer provided by the authors is very general and does not address my question regarding the sample. To reiterate: Is there z-drift induced by evaporation of the immersion water? And/or is there sample drift due to contraction/expansion of the hydrogel, or something else? I understand that the microscope setup will be subjected to some (small) drift due to the lack of temperature stabilization and vibration isolation in the room.

We apologize to the reviewer if our previous answer was too general and did not address the specific concern raised. In our work, we have mostly focused on correcting the drifts that we effectively observed during our acquisition process, rather than performing a detailed characterization of their potential sources. That being said, the main sources of drift in microscopy are commonly due to thermal fluctuations causing expansion/contraction of microscope components, as well as mechanical vibrations in the room (eg air con).

Concerning the possible z-drift induced by evaporation of the immersion water, we do not believe this is a significant source of drift as it would more likely lead to a sudden loss of immersion when the water layer becomes too thin. To avoid evaporation, we implemented a small water reservoir around the objective to maintain immersion throughout the acquisition. Other solutions, such as silicone-based immersion media with a refractive index similar to water, are now available and could also be considered in future implementations.

Regarding potential contraction or expansion of the polymer used in our SMART devices, we did not observe any noticeable impact. Indeed, such deformation would likely affect the quality or alignment of the light-sheet beam reflected by the 45° mirror — something we did not detect in our experiments.